# Nanocrystal Array Engineering and Optoelectronic Applications of Organic Small-Molecule Semiconductors

**DOI:** 10.3390/nano13142087

**Published:** 2023-07-17

**Authors:** Haoyu Gong, Jinyi Lin, Huibin Sun

**Affiliations:** Key Laboratory of Flexible Electronics (KLoFE), Institute of Advanced Materials (IAM), Nanjing Tech University, 30 South Puzhu Road, Nanjing 211816, China; 202061222181@njtech.edu.cn (H.G.); iamjylin@njtech.edu.cn (J.L.)

**Keywords:** organic small-molecule semiconductors, patterning, optoelectronic application, nanocrystal arrays

## Abstract

Organic small-molecule semiconductor materials have attracted extensive attention because of their excellent properties. Due to the randomness of crystal orientation and growth location, however, the preparation of continuous and highly ordered organic small-molecule semiconductor nanocrystal arrays still face more challenges. Compared to organic macromolecules, organic small molecules exhibit better crystallinity, and therefore, they exhibit better semiconductor performance. The formation of organic small-molecule crystals relies heavily on weak interactions such as hydrogen bonds, van der Waals forces, and π–π interactions, which are very sensitive to external stimuli such as mechanical forces, high temperatures, and organic solvents. Therefore, nanocrystal array engineering is more flexible than that of the inorganic materials. In addition, nanocrystal array engineering is a key step towards practical application. To resolve this problem, many conventional nanocrystal array preparation methods have been developed, such as spin coating, etc. In this review, the typical and recent progress of nanocrystal array engineering are summarized. It is the typical and recent innovations that the array of nanocrystal array engineering can be patterned on the substrate through top-down, bottom-up, self-assembly, and crystallization methods, and it can also be patterned by constructing a series of microscopic structures. Finally, various multifunctional and emerging applications based on organic small-molecule semiconductor nanocrystal arrays are introduced.

## 1. Introduction

Since the organic light-emitting diode (OLED) has become a research hotspot in academia and industry, organic small-molecule semiconductor materials have also come into people’s view. On the one hand, organic small-molecule semiconductor materials typically exhibit better semiconductor performance than polymer semiconductor materials due to their better crystallization properties. On the other hand, compared to traditional silicon-based semiconductor materials, organic small-molecule semiconductor materials also have the characteristic of being solution processable. Therefore, they are more suitable for the low energy consumption needs of future flexible electronic devices. In addition, the preparation of high-quality and high-density crystal arrays is crucial for the practical application of organic small-molecule semiconductor materials. Taking information display as an example, the density of an organic small-molecule crystal array determines the pixel density. Therefore, research on the processing technology of organic small-molecule semiconductors has always been a hotspot for organic electronics. However, organic small-molecule semiconductors are often subject to uncontrolled nucleation and growth during solution treatments, which results in poor reproducibility and performance of the nanocrystal arrays, limiting their function in electronic devices. To overcome this obstacle, considerable efforts have been put into adjusting the interface engineering conditions [1] for molecule stacking and nanocrystal array manufacturing by improving deposition techniques, solvent evaporation rates, liquid surface tension, heat treatment, and substrate surface energy. However, although considerable progress has been made in recent years, the correlation between preparation conditions and final morphology is still insufficient to explain the mechanism of solid solution treatment of organic small-molecule solutions [2,3]. Over the past few decades, various solution-based coating techniques [4] have been developed to control nanocrystal array crystallization, such as traditional spin coating [5]. At present, there are various methods for preparing organic small molecule crystal arrays [6], the main methods and means include the top-down method, bottom-up method [7,8], self-assembly method, crystallization method, and patterning method [9,10]. Top-down approaches often utilize micromachining methods [11] such as lithography. The bottom-up strategy uses controlled self-assembly, directed growth, and templates to prepare organic small-molecule semiconductor crystal arrays. Organic small-molecule semiconductors can also be used with inorganic semiconductors for high-performance transistors. In this paper, we will introduce the patterning of organic small-molecule semiconductor nanocrystal arrays and photoelectric applications and provide ideas for the future development of organic small-molecule semiconductor materials.

## 2. Methods

The methods of organic small-molecule semiconductor nanocrystal array are generally divided into spin coating and blade coating. In recent years, new methods of top-down, bottom-up, self-assembly, crystallization, and microstructure template have been developed, which greatly improve the photoelectric performance of organic small-molecule nanocrystal arrays, and can be improved on the basis of these methods by combining them with the use of microstructure templates to control the size and orientation of crystal arrays. These new methods can be used to prepare large scale organic small-molecule nanocrystal arrays with simple operation.

### 2.1. Spin Coating

Spin coating [12] is the most widely used solution processing method [13] in the field of organic electronics, which can be directly used in the preparation of organic field-effect transistor (OFET). The rate of rotation determines the rate at which organic small molecules crystallize and heal [14]. By slowing down the rotation rate of the substrate, this can further slow solvent evaporation and improve carrier mobility by orders of magnitude. For example, see the organic small-molecule semiconductor 6,13-bis(triisopropylsilylethynyl) pentacene (TIPS-PEN). Figure 1a shows the spin coating of TIPS-PEN nanocrystal arrays. Figure 1b shows the time evolution of Bragg slice strength (area under curve) at different rotational speeds (001). In Figure 1c, the optical microscope (OM) images of TIPS-PEN at different times can be seen. In Figure 1d, the relationship between film-forming time and rotation velocity is given. The disadvantage of spin coatings is that TIPS-PEN nanocrystal arrays contain many microcrystals with different orientations. Due to the action of centrifugal force, the thickness of the prepared organic small-molecule semiconductor nanocrystal arrays is not uniform.

### 2.2. Blade Coating

Nanocrystal arrays of organic small-molecule semiconductors, such as 2,8-difluoro-5,11-bis(triethylsilylethynyl)anthradithiophene (Dif-TES-ADT) are prepared using scraper coating technology and micro-groove writing devices. The scraper method is to place the organic small-molecule solution in the upper blade of the micro-groove, and then heat the bottom substrate. Using the shear force generated by the blade during scraping and coating, the continuous smooth organic small-molecule semiconductor nanocrystal arrays can be quickly prepared over a large area. The surface of the scraper has a very smooth blade, and speed can be applied to move it across the substrate, so that the solution of organic small molecules is evenly spread on the substrate and the nanocrystal array is formed. The charge mobility of the Dif-TES-ADT nanocrystalline array with an organic field-effect transistor (OFET) is 5.54 cm^2^V^−1^S^−1^. The difference between the direct writing method, the groove coating method, and the scraper method is that a tank containing solution is added above the blade, so that more organic small-molecule semiconductor nanocrystal arrays can be prepared. Figure 2a shows Dif-TES-ADT nanocrystal arrays that were prepared using the scraper method [15]. Figure 2b shows the groove coating method [16]. Figure 2c shows how the direct writing method is used to guide meniscus. The 6,13-bis(triisopropylsilylethynyl) tetraazapentacene (TIPS-TAP) casting is deposited in the scraper area [17]. In Figure 2d, when the micro-slot writer and the substrate move relative to each other, the 2-decyl-7-phenyl-[1]benzothieno[3,2-b][1]benzothiophene (Ph-BTBT-C10) solution is deposited on the surface [18]. The nanocrystal arrays prepared by these methods have uniformly oriented crystal bands and are simple to operate.

### 2.3. Microstructural Template Patterning Method

#### 2.3.1. Selective Contact Evaporation Printing

The selective contact evaporation printing (SCEP) method [19] can accurately prepare a variety of SC TIPS-PEN single-crystal arrays with the same molecular orientation, including square, rectangular, and hexagonal. This top-down etching technique, in which the size of the single crystal pattern can be controlled, uses the 6,13-bis(triisopropylsilylethynyl)pentacene (TIPS-PEN) molecule to undergo selective thermal evaporation at the polydimethylsiloxane (PDMS) interface of the elastomer, and then diffuses into the micro–nano structure of the mold to form a clear SC TIPS-PEN etching microregion. Figure 3a shows that SC TIPS-PEN was prepared using the SCEP method. Its switching current ratio and field-effect mobility are approximately 10^6^ and 0.36 cm^2^V^−1^S^−1^, respectively. A solution composed of small organic molecules flows outward under evaporation, transporting the solute to the edge of the droplet and causing the crystals to grow unevenly over a given area. Submicron scale high-fidelity 2,7-dioctyl[1]benzothieno[3,2-b] benzothiophene (C8-BTBT) single-crystal arrays are realized using the capillary force driven molecular flow selective contact method [20]. Figure 3b shows organic small-molecule C8-BTBT patterned nanocrystal arrays. Due to the vertical limitation of the deep channel mold, high-quality single crystal structure [21] is retained in the organic small-molecule nanocrystal arrays. Figure 3c shows optical microscopic images of microchannel C8-BTBT crystals at different temperatures. Figure 3d contains a scanning electron microscope (SEM) image of C8-BTBT crystal arrays. In this way, many arrays of organic small-molecule semiconductor nanocrystal arrays can be grown simultaneously. It can also control the crystal size by the size of the microstructure.

#### 2.3.2. Template-Assisted Self-Assembly (TASA)

Template-assisted self-assembly (TASA) technology [22] uses directional geometry to control the size and arrangement of nanostructures. It combines bottom-up self-assembly with top-down lithography template. Organic small-molecule semiconductor crystal materials can realize the ordered and regular arrangement of molecular structures in a semi-closed system, and it can be used for the patterning of organic small-molecule semiconductors [23]. Examples include C10-BTBT, TIPS-PEN, and TES-ADT. Figure 4a–d shows the template-assisted self-assembly method. The PDMS patterned template was impregnated with a 1,2-dichloroethane (DCE) solvent, then the template was placed on triethylsilylethynyl anthradithiophene (TES-ADT) nanocrystal arrays, and the rectangular pattern of TES-ADT nanocrystal arrays was obtained by separating the PDMS template [24]. This method can prepare organic small-molecule semiconductor crystal arrays with various patterns and shapes. The top contact of OFET with TES-ADT crystals exhibits an excellent field-effect mobility of about 0.3 cm^2^V^−1^S^−1^. Figure 4e–h shows the pre-template with polydimethylsiloxane (PDMS). The self-aligning properties of 2,7-didecylbenzothieno[3,2-b][1]benzothiophene (C10-BTBT) meso-crystalline liquid crystal phase enable it to be processed and micropatterned without solvent. The microstructure of organic small-molecule semiconductor crystal arrays can be regulated by controlling the evaporation of solvents within the dry droplet. In Figure 4i–l, planar-oriented crystal arrays with well-defined patterns and free surfaces can easily be obtained with the help of polyvinyl alcohol (PVA) micro-templates by simply melting meso-organic liquid crystal semiconductors at isentropic temperatures, cooling to a crystalline phase, and then sequentially removing the PVA templates. Figure 4m–p shows a polarization microscope (POM) image of an organic small-molecule semiconductor after removing the PVA template. The template-assisted self-assembly method has the advantage that it can be used for the preparation of many different kinds of organic small-molecule semiconductor nanocrystal arrays, the size of nanocrystal arrays can be controlled by designing the size of template microstructure, and it can also be applied on a large scale.

#### 2.3.3. Soft Photolithography

Soft photolithography has the advantages of short operation time [25], high yield, and multilayer stacking. It can reduce or increase the pattern of various nanocrystal arrays, including organic small-molecule semiconductors, luminescent chromophores, and metals, to produce complex and high-performance flexible electronic devices. Pattern pressure sensitive tape is used as the impression material for pattern transmission. In Figure 5a, by selectively separating or attaching a variety of organic small-molecule semiconductor nanocrystal arrays, its adhesion and flexibility allows the pressure-sensitive tape to form patterns on a variety of surfaces on organic polymer surfaces, inorganic surfaces, and bent and flexible substrates. There is asymmetric wettability of the microstructures [26] of 2,7-dioctyl[1]benzothieno[3,2-b] benzothiophene (C8-BTBT) single-crystal arrays with large areas of fine graphics that have been prepared with the characteristics of being highly crystalline, with crystal orientation, regular arrangement, and uniform size. Figure 5b shows that it has a hydrophobic side wall and a hydrophilic top, and uses the resulting capillary bridge to arrange microcolumns periodically so as to achieve unidirectional dehydration of organic solution and ordered accumulation of molecules, thus regulating mass transfer, molecular accumulation, nucleation, and crystal growth. One-dimensional organic single-crystal arrays have pure (100) crystal orientation, are π–π stacked in the optimal carrier transport direction, and have a carrier mobility up to 8.7 cm^2^V^−1^S^−1^. This method can improve the carrier mobility of organic small-molecule semiconductor nanocrystal arrays.

#### 2.3.4. Evaporative Assembly Method

Controlling the width of the microwires by controlling the amount of insulating polymer added can optimize the electrical properties of 6,13-bis(triisopropylsilylethynyl)pentacene (TIPS-PEN) organic small-molecule semiconductors. Ordered inorganic and organic microwire patterns are prepared using the top-down evaporation assembly method [27] for cross-stacking p-n heterojunction diode arrays [28]. N-type inorganic indium gallium zinc oxide (IGZO) microfilaments and P-type organic small-molecule TIPS-PEN microfilaments were treated with solution. In Figure 6a, polymethyl methacrylate (PMMA) solution is injected between the inclined metal blade and the IGZO substrate, trapping the solution through capillary forces. In Figure 6b, the non-volatile PMMA in the solution flows along the edge and migrates to the liquid–solid contact line driven by solvent evaporation. When the linear translational lateral movement on the substrate exceeds the set distance, the meniscus is stretched. When the contact line is moved to a new position, the contact angle returns to its initial value, resulting in PMMA microwires on the IGZO substrate. Figure 6c shows the top view of a cross-stacked p-n heterogeneous array junction diode. The width of the microfilaments is controlled by changing the interval stop time. A large number of organic small-molecule semiconductor nanocrystal arrays can be prepared using vaporization and self-assembly of the scraper, which has the characteristics of uniform orientation, low cost, and simple operation.

#### 2.3.5. Superhydrophobic Microcolumn Flow Coating Method

The superhydrophobic microcolumn flow coating (SMFC) method [29] allows the stacking, thickness, and position of crystals to be controlled by adjusting the flow rate, solvent hydrophobicity, and substrate hydrophobicity. It is also possible to control the height arrangement of the solution coating patterns by printing microcolumn patterns on the blades [30], increasing the size and spacing of the rectangular columns, thereby increasing the crystal size and thus achieving higher crystal array mobility. The 6,13-bis(triisopropylsilylethynyl)pentacene (TIPS-PEN) single-crystal arrays can be patterned using spatial constraints and field/force induction strategies. Its mobility is as high as 6.8 cm^2^V^−1^S^−1^. Figure 7a shows the preparation of TIPS-PEN nanocrystal arrays using the superhydrophobic microcolumn flow coating method. Figure 7b–e shows superhydrophobic microcolumns at different speeds: (b) 0.5 mms^−1^, (c) 1 mms^−1^, (d) 2 mms^−1^, and (e) 3 mms^−1^. There is an optical microscope image of prepared TIPS-PEN single-crystal array and corresponding atomic force microscopy (AFM) image. The more uniform oriented nanocrystal arrays are obtained by increasing the velocity, and the crystal array area is about 2 cm^2^. The microcolumn structure changes the curvature and density of the meniscus and can control the nucleation process. Figure 7f shows the solution shear process of microstructure allylhybridpolycarbosilane (AHPCS) leaves. Figure 7g shows a scanning electron microscope (SEM) image of the AHPCS microstructure shear blade. Figure 7h–j shows tha curvature of meniscus line formed between the blade and substrate. Figure 7l–p show cross-polarization optical micrographs (CPOMs) of TIPS-PEN nanocrystal arrays in different sizes of 10 μm, 15 μm, 20 μm, 40 μm, and 60 μm. Figure 7k shows the relationship between the mean crystal width of the tool tip and the microstructure size of the shear blade. It is the best method to manufacture low cost, large area, lightweight organic small-molecule semiconductor materials and field-effect transistor channel materials.

#### 2.3.6. Electro-Fluid (EHD) Jet Printing

Printing is also a method for preparing patterned arrays of two-dimensional organic small-molecule semiconductor crystals, such as towed electrohydrodynamic jet printing [31,32,33], roller pen writing [34], and direct writing [35]. Electrohydrodynamic (EHD) injection technology directly writes organic semiconductor crystals for patterning, enabling the preparation of large area organic small-molecule semiconductor crystal arrays without complicated processes. The drag mode provides favorable conditions for crystal growth by effectively controlling the power supply voltage and the distance from the nozzle to the substrate. It can produce a unidirectional array of nib crystals along the printing direction. Figure 8a shows a towed electrohydrodynamic writing 6,13-bis(triisopropylsilylethynyl)pentacene (TIPS-PEN) crystal array. In Figure 8b, high-performance and large area two-dimensional organic small-molecule semiconductor crystal arrays printed electronic products can be prepared by using a roller pen to write two-dimensional 2,7-dioctyl[1]benzothieno[3,2-b] benzothiophene (C8-BTBT) organic small-molecule semiconductor crystals directly. The obtained crystals have high crystallinity, good atomic finish, and large size. The average carrier mobility of the obtained OFET is 3.1 cm^2^V^−1^S^−1^ and the maximum carrier mobility is 5.92 cm^2^V^−1^S^−1^. The pattern of organic small-molecule semiconductor obtained by direct writing method is shown in Figure 8c. It is highly crystalline and purely oriented, showing higher device performance compared to the non-pure oriented crystal OFET. In Figure 8d, the TIPS-PEN solution ejected from the nozzle is dewy on the linear polyurethane acrylate (PUA) pattern, but wet on the bis(benzocyclobutene) BCB surface and has a higher energy than the PUA. These crystals are highly crystalline and favor uniform stacked structures for lateral charge transfer.

#### 2.3.7. Programmed Scraper Method

The programmed scraper coating technique [36] grows organic small-molecule semiconductor crystal patterns. Alternately defined low/high solution shear rates are programmatically used. The nib crystals grow in the low-speed region and form patterns in the high-speed region. Pattern crystal growth can be performed in the solvent wet/dehumidified region [37]. Polyurethane acrylate was selected as the dehumidification material; it has good hydrophobicity, and its pattern is easy to selectively grow on the precursor film. In Figure 9a, a uniaxial arrangement of 6,13-bis(triisopropylsilylethynyl)pentacene (TIPS-PEN) crystals was prepared using the programmed scraper coating method. Figure 9b shows the chemical structure of TIPS-PEN and polystyrene (PS). Figure 9c shows polarizing optical microscope (POM) images of TIPS-PEN crystal morphology at different shear rates. Figure 9d shows an optical microscope image of TIPS-PEN crystals of different widths growing on a PUA/SiO_2_ substrate. Figure 9e shows polarized optical microscopy (POM) images of TIPS-PEN crystals of different widths. Figure 9f shows atomic force microscopy (AFM) morphology of a TIPS-PEN crystal on polyurethane acrylate (PUA) substrate. In Figure 9g, the height of the cross-section is distributed. The OFET array is prepared by the method of programming control and crystal with different line spacing. By changing the programming parameters, a variety of crystal patterns can be created simply, demonstrating the wide availability of crystal patterns and printing techniques.

#### 2.3.8. Screen Printing

The screen printing method [38] can control the morphology and crystallinity of organic small-molecule semiconductor nanocrystal arrays to achieve high-performance organic optoelectronic devices, and it can prepare simple and effective channel-limited organic small-molecule semiconductor nanocrystal arrays, which have good crystallinity and preferred orientation. The printing process can be accomplished by dropping ink onto the mesh and moving the scraper to fill the mesh with ink [39]. Ink can penetrate the mesh and deposit on the substrate. Figure 10a shows the screen printing device. The crystals grow from both sides, meet and stop in the middle of the pattern, so that the growth axes of the organic small-molecule semiconductor crystal arrays are aligned with the coating direction. Figure 10b shows the deposition pattern of Polyvinyl alcohol (PVA) resist. Figure 10c shows the contact angle between water and PVA resist on 2,7-dioctyl[1]benzothieno[3,2-b] benzothiophene (C8-BTBT). Figure 10d shows an optical microscope image of PVA array with an area of 1 cm^2^ that is printed on C8-BTBT nanocrystal arrays. Figure 10e shows an enlarged image of PVA array on C8-BTBT. The insulating polymer in the mixed ink also helps to improve the crystallinity of the organic small-molecule semiconductor crystal arrays. The PVA resist is then screen printed on top of the organic layer and wet-etched or dry-etched to form the pattern, which results in the organic pattern being highly crystalline and having very similar crystal orientations. The screen scraping method [40] can improve the resolution of organic small-molecule semiconductor crystal arrays.

#### 2.3.9. Microchannel-Assisted Inkjet Printing (MA-IJP) Method

Microchannel-assisted inkjet printing (MA-IJP) [41] can be used to map arrays of organic small-molecule semiconductors with ordered crystal orientation. The micrometer-sized channel template can be used as a one-step method for capillary force to guide the wetting process of organic ink, and can also limit the dehumidification behavior caused by solution evaporation, so that the organic small-molecule semiconductor crystals can grow orderly over long distances. The patterned 2,7-dioctyl[1]benzothieno[3,2-b] benzothiophene (C8-BTBT) crystals exhibit a (010) crystal-oriented one-dimensional structure. The MA-IJP method is suitable for soluble organic small-molecule semiconductor based on pentacene derivatives, such as 6,13-bis(triisopropylsilylethynyl)pentacene (TIPS-PEN). In Figure 11a, C8-BTBT ink drops into the hydrophilic square area, under the action of capillary force along the microchannel wetting. In Figure 11b, COMSOL software is used to simulate the process of solution wetting. Figure 11c shows a cross-polarized optical microscope (CPOM) image of C8-BTBT nanocrystal arrays. Figure 11d shows that the CPOM images of C8-BTBT crystals in the same region all have polarization angles less than 45° and 0°. Figure 11e shows an atomic force microscopy (AFM) image of a C8-BTBT crystal in a microchannel. Figure 11f,g shows a scanning electron microscope (SEM) image of C8-BTBT nanocrystal arrays. The smooth cross-section of the crystal indicates good crystallinity. The MA-IJP strategy can obtain high quality crystal arrays and promote the development of integrated electronic devices. Microchannels can play a role in controlling wetting and dehumidification dynamics.

#### 2.3.10. Double Blade Coating Printing Technology

The two-blade coating printing technology [42] can be used for the patterning growth of organic small-molecule semiconductor nanocrystal arrays, which can limit the organic small-molecule semiconductor crystal arrays to a specific region on the water surface of the molecular plane, thus greatly reducing the number of nuclei, and it can control the water amount in the wet region by adjusting the coating speed [43]. At high coating rates, the viscous force exerted by the wet substrate dominates, which allows more water to adhere to the substrate and fill the wet region, which results in a 2,7-dioctyl[1]benzothieno[3,2-b] benzothiophene (C8-BTBT) mode with single crystal domains. The average mobility of a C8-BTBT single crystal is 11.5 cm^2^V^−1^S^−1^, which is much higher than that of the single-crystal film prepared using the conventional blade coating method. Figure 12a–c shows double-knife coating technology for the preparation of C8-BTBT nanocrystal arrays. Figure 12d,e shows cross-polarized optical microscope (CPOM) images of C8-BTBT nanocrystal arrays. Figure 12f shows a C8-BTBT nanocrystal array atomic force microscopy (AFM) image. In Figure 12g, one can count the number of crystal domains in each patterned region of the histogram. In Figure 12h, the distribution of C8-BTBT nanocrystal arrays in a hybrid can be observed by time-of-flight secondary ion mass spectrometry (TOF-SIMS). This can be widely used in other soluble organic small-molecule semiconductor materials, and provides a new idea for multi-component integrated electronic products.

### 2.4. Bottom-Up Method

#### 2.4.1. Straightforward Scooping-Up

The direct scooping (SU) method [44] was used to prepare compact and neatly arranged 6,13-bis(triisopropylsilylethynyl)pentacene (TIPS-PEN) single-crystal arrays, and its self-assembled large organic small-molecule semiconductor TIPS-PEN crystal array layers could be quickly generated. Higher crystal density can provide efficient charge transport in FET devices, with field-effect mobility up to 2.16 cm^2^V^−1^S^−1^. In Figure 13a, under the action of the directional capillary force generated on the surface of the crystal band, a crystal aggregate composed of multiple nematic crystals arranged side by side is formed. In Figure 13b, the TIPS-PEN crystal array strips are picked up directly. The direct shovel method is highly fluid, and it is an improvement on the Langmuir—Blodgett (LB) method, allowing for the assembly of highly ordered one-dimensional TIPS-PEN crystals. Direct scooping shortened the inter-crystal distance and can be easily transferred to other substrates.

#### 2.4.2. Patterned Microchannel Dip Coating (PMDC) Method

In the process of programmed impregnation, a crystal fringe array with controllable fringe spacing was prepared by adjusting the pulping rate of the substrate with low boiling point solvent [45]. It has a good molecular structure and high crystallinity, which is conducive to efficient charge transport. In Figure 14a, using a programming impregnated 6,13-bis(triisopropylsilylethynyl)pentacene (TIPS-PEN), the nib crystal forms micro streaks with defined gap intervals on the substrate. Figure 14b shows an optical microscope (OM) image and polarized optical microscope (POM) image of TIPS-PEN crystal. The patterned microchannel dip coating (PMDC) method [46] can control the growth position and direction of organic small-molecule semiconductor single-crystal arrays, which can realize the integration of high-performance organic circuits. It can solve the problem of the difficult-to-prepare crystal array patterns with high crystallinity and pure crystal orientation in traditional methods [47]. The PMDC method has good versatility and can be extended to a variety of soluble pentene materials, such as DIF-TES-ADT [48] and TIPS-PEN. Figure 14c shows the patterned microchannel dip coating (PMDC) method. Figure 14d shows a cross-polarized optical microscope (CPOM) image of a 2,8-difluoro-5,11-bis(triethylsilylethynyl)anthradithiophene (DIF-TES-ADT) crystal in a patterned PR microchannel template board. In Figure 14e, in the enlarged cross-polarized optical microscope (CPOM) image, DIF-TES-ADT crystals with free grain boundaries are highly arranged, covering the entire PR microchannel region. In Figure 14f, a scanning electron microscope (SEM) image of a DIF-TES-ADT strip crystal structure shows that it is well-arranged. Figure 14g. DIF-TES-ADT crystal arrays have jagged edges, depending on the nature of the organic small-molecule semiconductor. Figure 14h shows an atomic force microscopy (AFM) image and corresponding height distribution map.

#### 2.4.3. Meniscus Coating Method

The channel-restricted meniscus self-assembly strategy [49] solves the problem where the traditional solution method leads to the heterogeneous nucleation behavior and aggregation of polycrystalline molecules during crystal growth. PR channels were used to pattern the substrate at both micro- and nano-scales to constrain the meniscus [50,51,52,53], thereby reducing the size of the anterior meniscus. In Figure 15a, when the PR channel is immersed in organic solution, a meniscus contour contact line is formed between two adjacent PR fringes. Figure 15 shows the C8-BTBT crystal array growth process in the channel-restricted meniscus. Figure 15c shows that the nucleation of organic small-molecule solutions along the leading edge of the meniscus is uniform. The 2,7-dioctyl[1]benzothieno[3,2-b] benzothiophene (C8-BTBT) solution is dragged onto the surface of the poly(4-vinylphenol) (PVP) pattern [54]. A large amount of the solution is confined between the silicon blade and the substrate by exposing the curved liquid level at the front end. As the solvent evaporates, the molecules selectively nucleate and crystallize on the surface of the PVP pattern, forming highly ordered C8-BTBT crystal arrays in the PVP-treated region. Figure 15d shows an atomic force microscopy (AFM) image of the C8-BTBT crystal surface after processing flat. Figure 15e shows that C8-BTBT crystal’s surface is clear, with high uniformity. Figure 15f is a cross-polarization optical microscope (CPOM) image. Microchannels can produce a channel-limiting effect, which can restrict the channel of the meniscus, so that the organic small-molecule semiconductor crystal is uniformly nucleated in the front of the meniscus. When the impregnated coating is pulled upward, the crystal growth direction can be consistent, thus giving the crystal better electrical properties.

### 2.5. Crystallization Dynamic Control

Organic small-molecule semiconductors are generally insoluble in inorganic solvents such as water, but easily soluble in organic solvents such as toluene and chlorobenzene to form organic solutions. Inorganic solvents such as water can be used as liquid soft templates for the growth of organic small-molecule semiconductor crystal arrays. Organic solvents such as chlorobenzene are easily volatile, so they can grow in solid microstructural templates. This can be changed by changing the temperature to make the crystals grow faster, and can improve electrical performance. The microstructure template can control the shape and size of crystal growth, which is a new crystal growth method.

#### 2.5.1. Solvent Crystallization Resistance Technology

The antisolvent crystallization technology of inkjet printing [55] can be used to prepare highly crystalline organic small-molecule semiconductor nanocrystal arrays [56]. It can also can control the growth of uniform single-crystal or polycrystal crystal arrays at the liquid–vapor interface, and obtain thin crystal arrays with an average carrier mobility of 16.4 cm^2^V^−1^S^−1^. Antisolvent crystallization is the best method to achieve controllable and expandable curing. The ink used is mixed with the antisolvent, and the ink can be printed individually at any location, forming a micro-liquid mixture between the inks on top of the substrate. It can solve the problem of uneven film thickness distribution caused by the traditional inkjet printing process, and it is an extension of double-head inkjet printing [57,58] for the preparation of organic small-molecule semiconductor nanocrystal arrays [59]. Figure 16a,b shows the process of preparing C8-BTBT crystal arrays using solvent-resistant ink crystal printing. Figure 16c shows an orthogonal Nichols polarizing optical micrograph of nanocrystal arrays. In Figure 16d is the magnification of the microscopic streaks of the crystal arrays. Figure 16e shows an atomic force microscopy (AFM) image and height profile. This printing technology is an important step toward high-performance single-crystal semiconductor devices for large-area and flexible electronic applications.

#### 2.5.2. Temperature Gradient Technique

The organic small-molecule semiconductor nanocrystal arrays grown using the drip casting method have random orientation and a poor coverage area, which will lead to poor performance between organic field-effect transistor (OFET) devices, when using, for example, 6,13-bis(triisopropylsilylethynyl) pentacene (TIPS-PEN), etc. Temperature gradient technology [60] can solve these problems [61]. The hole mobility of a TIPS-PEN organic field-effect transistor (OFET) was improved by the addition of poly(a-methyl styrene) PαMS polymer and the combination of a temperature gradient technique [62]. Figure 17a–d shows that when the solution reaches supersaturation, TIPS-PEN crystals begin to grow from the cold zone to the warm zone. The addition of the PαMS polymer eliminates the hot cracking phenomenon in the crystal arrays, and it makes the homogeneity orientation of the crystal more regular. In the digital image in Figure 17e, PαMS/TIPS-PEN nanocrystal arrays without thermal cracks were prepared using the temperature gradient method, which overcame the disadvantage of small coverage area of random orientation. Compared with the polymer without PαMS, the decrease in crystal width and the presence of PαMS polymer matrix can effectively relieve the thermal stress during the crystallization process and prevent the formation of thermal cracks.

#### 2.5.3. PDMS-Assisted Crystallization Method

Using the polydimethylsiloxane (PDMS)-assisted crystallization method, it is possible to make the 6,13-bis(triisopropylsilylethynyl)pentacene (TIPS-PEN) crystal arrays with a certain in-plane orientation, which is similar to the type I phase lattice structure and in-plane lattice orientation arrangement. Figure 18a shows a PDMS-assisted crystal growth TIPS-PEN crystal array. Figure 18b–e shows polarization optical microscope (POM) images of a TIPS-PEN crystal array. The adjacent blue and yellow crystals in Figure 18f–i show double crystals with (010) faces as double borders. In the nucleation stage of TIPS-PEN crystals, the A-axis of adjacent crystals can be aligned along the growth direction, but the B-axis direction cannot be adjusted. If the TIPS-PEN nucleation area is treated with a hydrophilic surface treatment, the formation of the twin boundary can be reduced. In Figure 18j, the TIPS-PEN crystals are arranged roughly along the top right and bottom left directions. In order to obtain higher charge mobility, the PDMS-assisted crystallization method arranges the crystals in a plane direction to reduce defects [63]. If a solvent with a high boiling temperature is used, the out-of-plane orientation of the TIPS-PEN molecules shows consistent growth [64], and it reduces the production of low angle grain boundaries (LAGBs). The improved out-of-plane lattice arrangement can make the performance of the device more stable.

#### 2.5.4. Heat-Induced Self-Assembly Method

Heat-induced self-assembly technology can be used for ultra-thin organic small-molecule semiconductor crystal arrays with a certain number of layers and millimeter-level coverage [65]. It is a basic method of molecular layer coating precisely defined by interfacial interactions, and it can produce complex, high-performance, and large area organic electronic devices. Monolayers are used as templates for the epitaxy of highly uniform N-type semiconductor, resulting in heterojunctions with significant bipolar charge transfer behavior. The annealing temperature can be increased to precisely control the accumulation of molecules at the interface [66], forming a large area of regular two-dimensional layered molecular crystal arrays instead of equilibrium rod-like crystal arrays, which will facilitate the precise control of 2,7-dioctyl[1]benzothieno[3,2-b] benzothiophene (C8-BTBT) nanocrystal arrays as bilayer (2L) or single-layer (1L) molecular nanocrystal arrays [67]. Figure 19a shows heat-induced self-assembled C8-BTBT nanocrystal arrays. Figure 19b–e shows the atomic force microscopy (AFM) and HRAFM images of double-layer and single-layer C8-BTBT nanocrystal arrays. In Figure 19f–h, when the single-layer diameter (D1L) is 1.6 mm, the coverage is maximal. In Figure 19i, the escaped molecules are self-assembled to flip over to a minimum of 1L. Figure 19j shows the adhesion curves of 1L C8-BTBT nanocrystal arrays and SiO_2_, 2L C8-BTBT, and 1L C8-BTBT. In Figure 19k, the assembly length of C8-BTBT nanocrystal arrays varies with substrate temperature, and three layers (3L) and thicker layers are difficult to obtain by adjusting the annealing temperature.

#### 2.5.5. Two-Step Crystallization Process

A high-quality single-crystal p−n heterojunction (SCHJ) was prepared using orthogonal solvent two-step crystallization [68], which is an interfacial solution crystallization process [69]. In Figure 20a, a second layer of single crystals is formed at the interface of the first layer of single crystals. During the crystallization of the second layer, the first layer is destroyed, which is the problem to be solved in the second stage. A single 6,13-bis(triisopropylsilylethynyl)pentacene (TIPS-PEN) crystal can grow without dissolving the first layer by choosing 4-methyl-2-pentone, which can dissolve TIPS-PEN effectively but cannot dissolve $C_{60}$ as a solvent [70], and the substrate can be annealed under vacuum to remove the residual solvent. Figure 20b shows a scanning electron microscope (SEM) image of the TIPS-PEN nanocrystal array. Figure 20c shows energy dispersive spectroscopy (EDS)-related element mapping. Figure 20d shows the energy dispersion spectrum of the crystals. In Figure 20e, the thickness of the overlapping SCHJ was studied using atomic force microscopy (AFM). In Figure 20f, the samples were screened using transmission electron microscopy (TEM). Figure 20g shows how the crystallographic analysis of the bilayer material was carried out by select area electron diffraction (SAED). The double-layer SAED shows two sets of diffraction points (blue and yellow circles), representing two different single crystals. This method is beneficial to the preparation of a heterojunction.

#### 2.5.6. Micro-Spacing Air Sublimation Method

The micro-spacing air sublimation method [71] can pattern small organic molecule single crystals, and the method of air sublimation can assist the solid substrate surface to perform wettability treatment [72], which can precisely control the growth position of an organic small-molecule semiconductor, so that the interconnected single crystal patterns have uniform crystal orientation, and higher crystallinities. Using, for example, 2,7-dioctyl[1]benzothieno[3,2-b] benzothiophene (C8-BTBT), this method can be used to prepare single crystal patterns of different shapes and sizes with uniform orientation. Its mobility is 6.28 cm^2^V^−1^S^−1^. This method overcomes the uncontrolled growth of isolated crystal patterns on non-epitaxial substrates and enables the arrangement of anisotropic electron single crystals to be integrated in large-scale devices. Figure 21a ahows the photolithography patterning of wettable substrates. Figure 21b shows optical microscope (OM) and scanning electron microscope (SEM) images of C8-BTBT crystals of different shapes and sizes. The pattern has regular shapes and sharp edges. The isotropic C8-BTBT melt crystallizes continuously from the intermediate liquid crystal state, and a uniform oriented patterned crystal array can be obtained [73].

#### 2.5.7. Vapor-Induced Coating Method

Organic small-molecule semiconductor 2,7-dioctyl[1]benzothieno[3,2-b] benzothiophene (C8-BTBT) striped single crystals with good orientation were prepared using vapor-induced coating [74,75]. The coating rate and solution concentration can control the morphology of the striated crystals, thereby changing the wetting behavior of the three-phase contact line and thereby altering the mass transport of the meniscus. The lack of a solute supply leads to the formation of dendritic crystals. Figure 22a shows uniformly striped crystals of C8-BTBT. Vapor molecules diffuse from the vapor source into the air in a concentration gradient, forming a Marangoni flow that guides the solution through dehumidification. In Figure 22b, the morphology of C8-BTBT nanocrystal arrays was obtained under different surface tensions and substrate wettability. Figure 22c shows layer-controlled C8-BTBT coating. The figure shows the change h/d of the steam source system. Figure 22d shows each stage of steam-induced coating process. Figure 22e shows the simulation results of vapor molecular distribution, velocity distribution, and solute concentration in the meniscus during the coating process. Different crystal array heights and solute concentrations determine the total solution mass at the crystallization point [76], which make it possible to control the formation process of molecular scale crystal arrays from single layer to multilayer.

### 2.6. Self-Assembly Method

#### 2.6.1. Van Der Waals Force Epitaxial Growth

The van der Waals epitaxy method [77] can be used to grow single or multilayer organic small-molecule semiconductor single-crystal arrays on graphene or boron nitride (BN) substrates on a large scale, which can precisely control the morphology of two-dimensional layered organic small-molecule semiconductor materials [78]. The weak van der Waals interaction makes the surface of the crystal layer very smooth, which is conducive to the complete separation and transfer of the crystal array from the substrate. The initial phase of epitaxy depends on the combination of the van der Waals forces between the molecule and the substrate, which include the flatness of the substrate [79]. The carrier mobility of a single-layer C8-BTBT organic small-molecule semiconductor FET can reach 10 cm^2^V^−1^S^−1^. This new type of two-dimensional molecular material plays an important role in the field of optoelectronics. Figure 23a shows the structural arrangement of C8-BTBT. Figure 23b–e shows atomic force microscope (AFM) images of a C8-BTBT crystal at different growth stages. Figure 23f–k shows the epitaxial growth of C8-BTBT molecular crystals on graphene, with an initial two layers (interfacial layer IL and first layer 1L). In each crystal layer, growth begins at the nucleation position and proceeds in an isotropic, tightly packed fashion. Figure 23l–o shows optical microscopy and cross-polarized light microscopy of C8-BTBT nanocrystal arrays. Two-dimensional molecular crystals are widely used in layered materials and heterogeneous structures. The C8-BTBT molecular crystal epitaxial growth method has strong processability and application value.

#### 2.6.2. External Force Driven Solution Epitaxy (EFDSE) Method

The solution epitaxy driven by external force can grow high-quality and high-performance [80] two-dimensional organic small-molecule semiconductor nanocrystal 2DOSC at the air/water interface [81], using, for example, 2,7-didecylbenzothienobenzothiophene (C10-BTBT). The surface flow generated by the external force can maintain the stable diffusion of the organic solution, so as to ensure the directional two-dimensional structure of continuous epitaxial growth of crystal and achieve the best charge transfer. In Figure 24a, C10-BTBT solution can be diffused and grown by the external force generated when an octadecyltrichlorosilane (OTS)-modified slider is inserted into water. S represents the diffusion capacity of the solution on the water surface, as shown in Figure 24b. S = $\gamma_{1}$ − $\gamma_{2}$ − $\gamma_{12}$ ($\gamma_{1}$ and $\gamma_{2}$ are the surface tension of the water and the solution respectively, and $\gamma_{12}$ is the surface tension of the interface between the two solutions). When S > 0, the solution diffuses on the water surface, forming a highly arranged and large area of the 2DOSC growth mechanism [82]. If the surface tension of the solvent is much less than the surface tension of water (such as ethyl acetate), the solution of organic small-molecule semiconductor disperses too quickly in water without enough time for C10-BTBT 2DOSCs to grow continuously, with only a few sheets scattered randomly on the water surface. If S < 0, the solution will remain at the interface and will not disperse (chlorobenzene). The insertion of the slide must control the steady motion of the contact line, which is conducive to the growth of a large area of 2DOSCs.

#### 2.6.3. Self-Assembly Induced by Graphene Quantum Dots (GQDs)

The solution-induced graphene quantum dots (GQDs) self-assembly strategy [83] can solve the problem of heterogeneous diffusion of organic solution on water’s surface and the interaction between organic small-molecule semiconductors to grow a single layer of highly crystalline C10-BTBT nanocrystal arrays on water’s surface. It combines the advantages of solution self-assembly and graphene-induced epitaxial growth, using GQD solutions to enhance the diffusion ability of organic solutions over water. In Figure 25a–c, adjusting the pH value of GQD solution can reasonably control the diffusion area and promote the growth of monolayer organic crystals. In Figure 25d,e, the π–π stacking interaction between GQDs [84] and organic molecules can significantly reduce the interaction force of organic molecules and make it easier for organic small-molecule semiconductor crystal arrays to join together, which allows monolayers to grow over a large area, which opens up a new method for the preparation of large area organic monolayers.

#### 2.6.4. Soft Template Assisted Self-Assembly

The soft template assisted self-assembly (STAS) strategy [85] uses liquid substrate as soft template to grow 2D organic small-molecule semiconductor crystals (2DMCs) with different structures in solvents with different surface tension, which provides a new method for the growth of high-performance organic small-molecule semiconductor crystals. Adding a solvent with high surface tension in water can achieve uniform diffusion of a good organic small-molecule solution, and 2DMCs can be quickly transferred layer by layer. The surfactant potassium perfluorooctane sulfonate $C_{8}F_{17}{KO}_{3}S$ was added to the aqueous phase to form a molecular layer as a soft template. Amphiphilic surfactant molecules spontaneously aggregate at the air/water interface to form hydrophilic groups and hydrophobic groups, which can be used as the soft molecular layer interface layer. This is conducive to the continuous diffusion of thin solution layers on the water’s surface, which can form a large two-dimensional molecular crystal layer. In Figure 26a–i, C6-DPA toluene solution grows two-dimensional organic single-crystal arrays on the surface of different concentrations of surfactants. The solvent with low surface tension or high surface tension can be continuously diffused in water without splitting, which is conducive to the spreading of ultra-thin large area 2DMC arrays.

#### 2.6.5. DOSC Two-Step Strategy

In a two-step growth strategy for two-dimensional organic small-molecule semiconductor crystal arrays (2DOSCs) in patterned high-resolution layers, large-scale two-dimensional organic small-molecule semiconductor crystal arrays with controllable molecular layers are first prepared on a glycerol substrate and transferred to the target substrate [86]. The high-resolution laminated 2DOSC array was obtained using the selective contact evaporation printing (SCEP) technique with the help of the PDMS template, which can realize the highly integrated two-dimensional organic small-molecule semiconductor crystal array using, for example, 2,6-bis(4-hexylphenyl)anthracene (C6-DPA). In Figure 27a, the two-step strategy of the 2DOSC array starts by controlling the concentration of C6-DPA solution [87,88] on the glycerol substrate, which can obtain C6-DPA from monolayer (1L) to multilayer crystal arrays. There is selective etching of C6-DPA 2DOSCs at the PDMS/2DOSC interface using the thermal evaporation method. As can be seen in Figure 27b,c, this strategy can obtain a large area C6-DPA 2DOSC array image. It can prepare 2DOSC patterned crystals on a large scale with simple operation and low cost.

## 3. Photoelectric Application

### 3.1. Artificial Neural Network Synapses

The dielectric surface of the organic small-molecule semiconductor C8-BTBT phototransistor depends on the broad band optical response behavior. The recognition rate using SiO_2_-based OPT to simulate photoelectric artificial synapses [89] is over 95%, while the CYTOP-based OPT has a recognition rate of 0% when using the same semiconductor. The optical reaction in the 400–1000 nm region depends on the optical gating effect on the surface of the medium, which is a band with less energy than the 2,7-didecylbenzothienobenzothiophene (C8-BTBT) band gap. At 400 nm, the Vturn shift of SiO_2_-based OPTs is up to 43 V, the sensitivity >1.74 × $10^{4}{A\cdot A}^{-1}$, and the detection rate >2.40 × $10^{12}$ Jones. Artificial synapses [90,91,92,93] are simulated to realize face pattern recognition [94]. In OPT based on hydrophilic media, artificial synapses can recognize an ultra-wide optical range from ultraviolet (UV) to near-infrared (NIR). Figure 28a shows signal transmission between input and output neurons. In Figure 28b–d, in the simulation of synaptic neuro-morphism, a fully connected bilayer neural network consisting of 32 × 32 input neurons, 20 intermediate layer neurons, and 3 output neurons was designed. Figure 28e–g shows enhancement and inhibition (P–D) curves of OPT based on SiO_2_. Figure 28h–j shows the OPT enhanced inhibition (P–D) curve based on CYTOP. Based on these results, organic small-molecule semiconductor crystal arrays can be applied to artificial neural networks (ANNs) for pattern recognition, and the photogating effect has a strong photoelectric performance in neuromorphic computing systems. The organic small-molecule phototransistor has good visual recognition performance and strong practicability.

### 3.2. Organic Photodetector

Organic photodetectors have high application value in environmental health monitoring, quantum communication [95], chemical/biomedical sensors, image sensors, and other fields. Organic phototransistor (OPT) channel carriers are composed of electrons and photons, and they have higher photosensitivity and lower noise than ordinary diode photodetectors [96]. We analyzed a high-performance dual-band phototransistor with ${CH}_{3}{NH}_{3}P_{b}I_{3}$ nanoparticles (NPs) coated with C8-BTBT single-crystal array hybrid structure. Compared with the absorption of C8-BTBT and ${CH}_{3}{NH}_{3}P_{b}I_{3}$ in ultraviolet (UV) and visible (vis) regions, the hybrid structure of C8-BTBT and ${CH}_{3}{NH}_{3}P_{b}I_{3}$ shows the absorption covering the whole UV–visible range and has the best photoelectric detection performance. Hybrid ${CH}_{3}{NH}_{3}P_{b}I_{3}$/C8-BTBT single-crystal array integrated phototransistor circuits are widely used in high security communication. Figure 29a–g shows the detection capability, transmission characteristics, optical responsiveness, and energy level characteristics of a hybrid photodetector composed of a C8-BTBT single-crystal array.

### 3.3. DPA SC Polarized Light Communication System

The bridge LIB strategy between the organic small-molecule semiconductor single-crystal emitter and liquid insulator of DPA SC results in highly efficient structured LP-LED [97], and the DPA SC with high mobility and strong luminescence characteristics is used as the active emission layer to improve the transmission performance of injected carriers. This promotes the radiation coupling of excitons. It overcomes the common problem of the short circuiting of upper and lower electrodes in traditional devices, as well as the problem of cracking of the upper electrode caused by an ultrathick SC. LP-LED of DPA SC shows excellent luminous performance. Its maximum brightness and EQE are 2427 cdm^−2^ and 3.38%, respectively. The highly polarized light emission of an organic small-molecule semiconductor single-crystal LP-LED is caused by the generation of excitons between anisotropic molecules and different resonances along the crystal axis. High-efficiency LP-LED is a new type of polarized light communication system between chips. It surpasses traditional materials and methods in high-performance organic SC LP-LED polarizing and optoelectronic device applications. The system can detect the anisotropy of multi-dimensional information and detect the polarization signal. Figure 30a–e shows the polarized optical communication system between DPA SC chips and working parameters.

### 3.4. Photoelectric Storage

High photosensitive organic small-molecule semiconductors can produce photoelectric LIM units with multiple layers of storage [98]. The biggest advantage of photoelectric memory is the security of data storage [99]. In conventional memory, data are mainly recovered by applying a bias voltage. Optical control storage information can only be obtained by applying specific wavelength/intensity optical signal, thus ensuring its security. Organic small-molecule semiconductor C8-BTBT nanocrystal arrays are highly responsive to ultraviolet (UV) optical signals, while PhC2H4-benzo[de]isoquinolino[1,8-gh]quinolone diimide (PhC2-BQQDI) transistors are photo-responsive to visible light. Figure 31a–f shows the operating principle of ternary photoelectric storage. The optical response of C8-BTBT and PHC2-BQQDI organic small-molecule semiconductor transistors enables visible-induced programming and UV-induced erasure operations. Using a p-type C8-BTBT transistor and n-type PhC2-BQQDI transistor, the binary storage inverter is realized. The organic MVL’s LIM unit achieves high data integration, photoelectric storage operation, and multilevel storage [100]. Memory logic (LIM) integrates logic and memory operations into a single device architecture to achieve the functionality of a ternary LIM.

## 4. Conclusions

Processable organic small-molecule semiconductors are promising candidates for FET channel materials of low cost, large area, and light weight, with flexible electronic applications. In order to control the formation of organic small-molecule nanocrystals, a variety of solution-based coating technologies have been developed. At present, new methods such as top-down, bottom-up, self-assembly, crystallization, and visualization have been developed to solve the preparation of organic small-molecule semiconductor crystal arrays. During solution treatment, organic small-molecule semiconductors are often subject to uncontrolled nucleation and growth, which results in poor reproducibility and performance of the crystal arrays, limiting their function in electronic devices. Many efforts have been made to overcome this obstacle by improving and adjusting the molecular stacking conditions or through interface engineering. Constructing a variety of microstructure patterns to grow organic small-molecule semiconductor nanocrystal arrays is a new solution. The size of organic small-molecule semiconductor crystal arrays is usually 1–10 cm^2^, which greatly improves the photoelectric performance of organic small-molecule semiconductor nanocrystal arrays and can play an important role in photodetectors, artificial synapses, and photoelectric storage.

## Figures and Tables

**Figure 1 nanomaterials-13-02087-f001:**
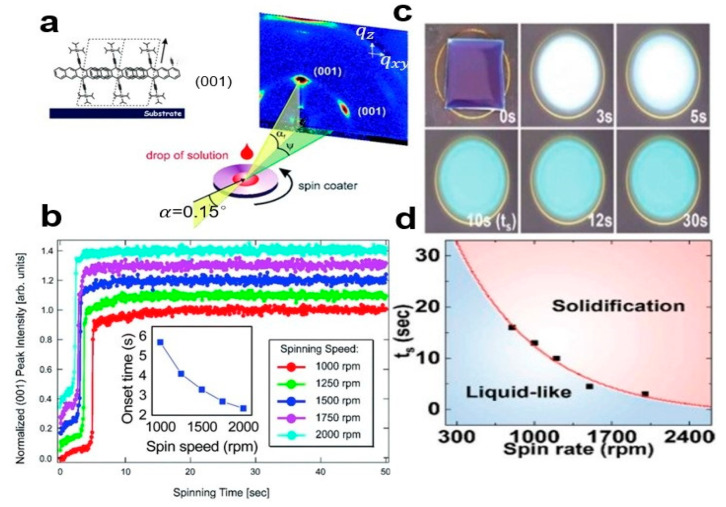
(**a**) The (001)-oriented TIPS-PEN nanocrystal array was obtained using the spin coating method. (**b**) Absorption strength of TIPS-PEN crystals at different rotational speeds [12]. Copyright© The Royal Society of Chemistry 2014. (**c**) Optical microscope (OM) images of TIPS-PEN crystals at different times. (**d**) The relationship between growth time and rotation rate of crystal arrays [13]. Copyright© The Royal Society of Chemistry 2019.

**Figure 2 nanomaterials-13-02087-f002:**
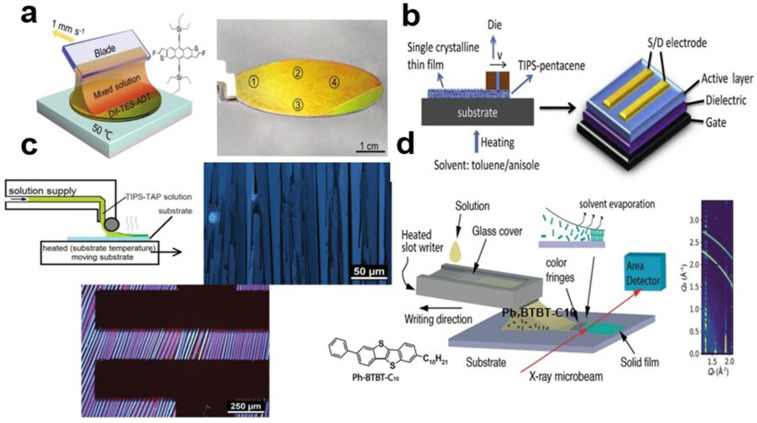
(**a**) Scraper method [15]. Copyright© The Royal Society of Chemistry 2020. (**b**) TIPS-PEN was prepared by trough coating [16]. Copyright© 2017 Elsevier B.V. All rights reserved. (**c**) Regional casting deposition of TIPS-TAP solution [17]. Copyright© The Royal Society of Chemistry 2016. (**d**) Micro-slot writing of Ph-BTBT-C10 solution [18]. Copyright© The Royal Society of Chemistry and IChemE 2022.

**Figure 3 nanomaterials-13-02087-f003:**
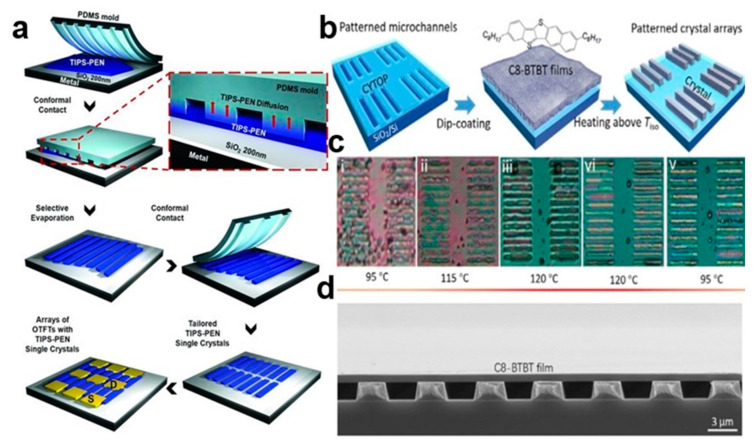
(**a**) SCEP method [19]. Copyright© 2011 WILEY-VCH Verlag GmbH & Co. KGaA, Weinheim. (**b**) C8-BTBT microstructured crystal array prepared using capillary force driven molecular flow [21]. (**c**) Optical microscopic images of microchannel C8-BTBT crystals at different temperatures. (**d**) SEM image of C8-BTBT crystal arrays. Copyright© 2020 Elsevier Ltd. All rights reserved.

**Figure 4 nanomaterials-13-02087-f004:**
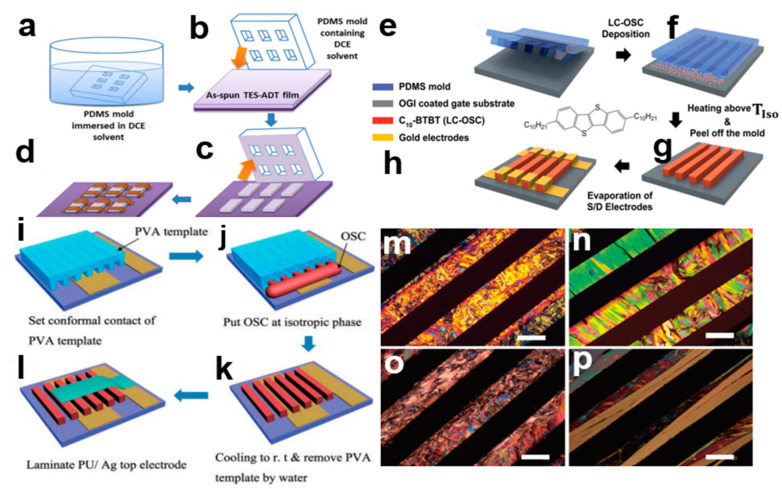
(**a**) The PDMS template is dipped into 1,2-dichloroethane (DCE) solvent. (**b**) The DCE wet PDMS template is placed on the TES-ADT nanocrystal arrays. (**c**) The PDMS template is separated to obtain the TES-AD crystal pattern. (**d**) The Au electrode and TES-ADT constitute the OFET device [23]. Copyright© 2013 WILEY-VCH Verlag GmbH & Co. KGaA, Weinheim. (**e**) The PDMS template begins to contact the ITO substrate. (**f**) C10-BTBT powder is placed into the channel. (**g**) At a certain temperature, the liquid C10-BTBT crystal material fills in the groove by capillary action, and after separating the PDMS template the C10-BTBT crystal pattern is obtained. (**h**) C10-BTBT can be used as a photoelectric device [24]. Copyright© 2016 Elsevier B.V. All rights reserved. (**i**) Planar-oriented organic small-molecule semiconductor crystal PVA template. (**j**) PVA template channels are added to organic small-molecule semiconductors. (**k**) The PVA template is separated from the crystal array. (**l**) Preparation of organic small-molecule semiconductor devices. (**m**–**p**) Polarizing microscope images: C10-BTBT, C8-BTBT, C4-BTBT, TTP [22]. Copyright© 2017 The Japan Society of Applied Physics.

**Figure 5 nanomaterials-13-02087-f005:**
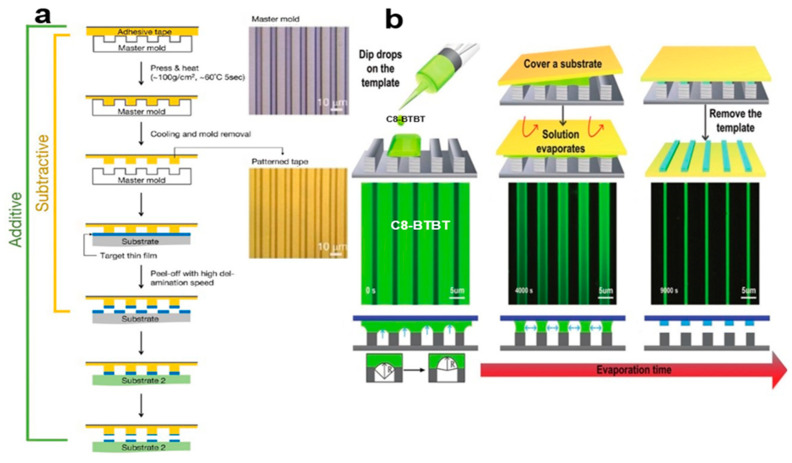
(**a**) Organic small-molecule crystal arrays using the soft lithographic pattern method [25]. Copyright© 2016 American Chemical Society. (**b**) In situ fluorescence microscope image of one-dimensional C8-BTBT single-crystal strip array [26]. Copyright© 2018 WILEY-VCH Verlag GmbH & Co. KGaA, Weinheim.

**Figure 6 nanomaterials-13-02087-f006:**
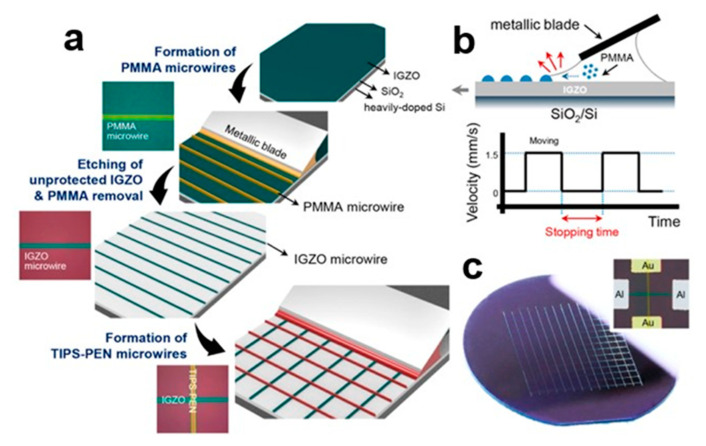
(**a**) The p-n heterojunction of TIPS-PEN and IGZO cross-stacked microfilaments prepared using evaporation and self-assembly [28]. (**b**) Metallic blade movement speed and time relation graph. (**c**) Nanocrystal array image of p-n heterojunction. Copyright© 2016 American Chemical Society.

**Figure 7 nanomaterials-13-02087-f007:**
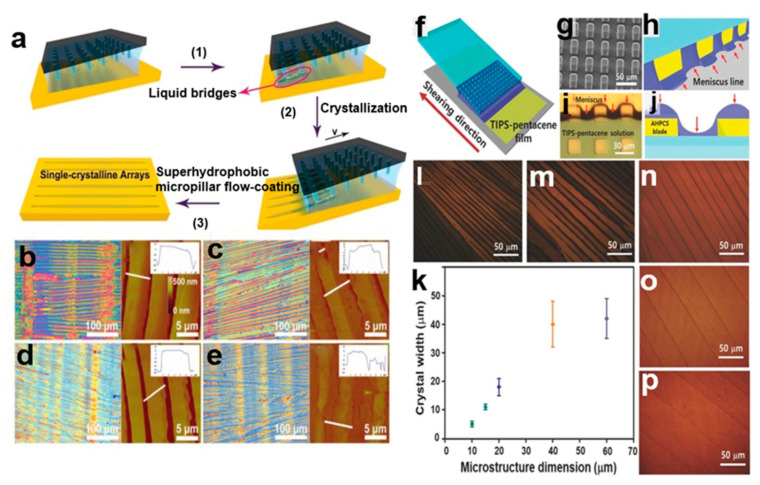
(**a**) TIPS-PEN nanocrystal arrays created using the SMFC method [29]. (**b**–**e**) Optical microscope images and corresponding AFM images of TIPS-PEN single-crystal array with superhydrophobic microcolumns at different velocities: (**b**) 0.5 mms^−1^, (**c**) 1 mms^−1^, (**d**) 2 mms^−1^, (**e**) 3 mms^−1^. Copyright© The Royal Society of Chemistry 2017. (**f**–**j**) Solution shear process of microstructured AHPCS leaves [30]. (**l**–**p**). Cross-polarization optical micrographs of TIPS-PEN nanocrystalline arrays of different sizes: (**l**) 10 μm, (**m**) 15 μm, (**n**) 20 μm, (**o**) 40 μm, (**p**) 60 μm. (**k**) The relationship between the mean crystal width of the tool tip and the microstructure size of the shear blade. Copyright© 2018 WILEY-VCH Verlag GmbH & Co. KGaA, Weinheim.

**Figure 8 nanomaterials-13-02087-f008:**
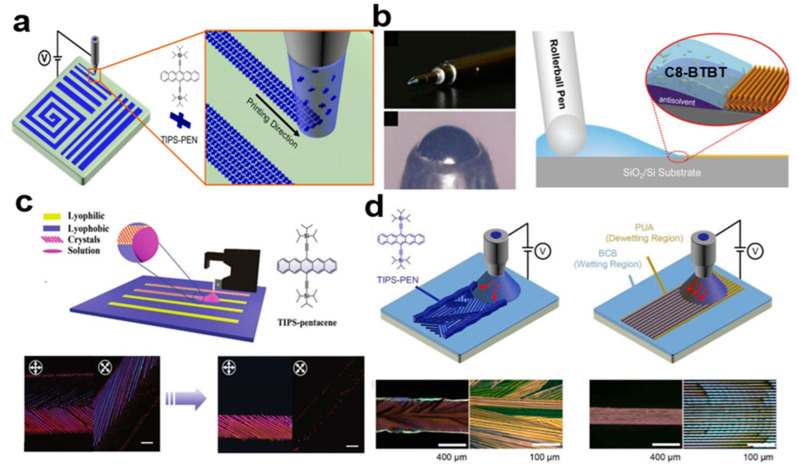
(**a**) Electrohydraulic dynamic injection TIPS-PEN nanocrystal arrays [32].Copyright© 2017 American Chemical Society. (**b**) Preparation of C8-BTBT nanocrystal arrays using the pen rolling technique [34]. Copyright© The Royal Society of Chemistry 2017. (**c**) Direct writing TIPS-PEN nanocrystal arrays [35]. Copyright© 2022 Wiley-VCH GmbH. (**d**) Polarized optical microscopy (POM) images of TIPS-PEN crystals prepared using the EHD method [31]. Copyright© 2022 Elsevier B.V. All rights reserved.

**Figure 9 nanomaterials-13-02087-f009:**
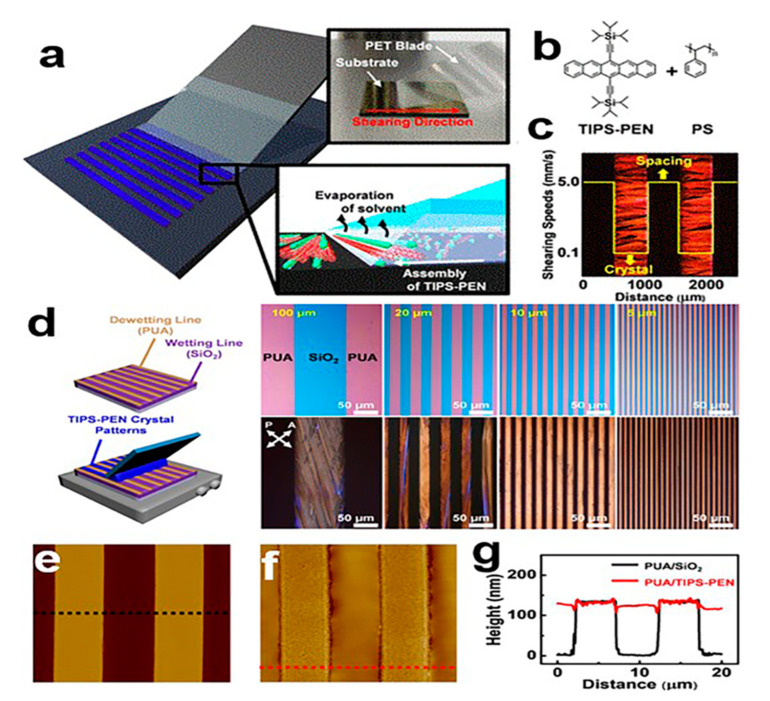
(**a**) Programmed scraper coating method [36]. (**b**) Chemical structure of TIPS-PEN and polystyrene (PS). (**c**) Polarizing microscope (POM) images of TIPS-PEN crystal patterns at different shear rates. Copyright© 2019 American Chemical Society. TIPS-PEN nanocrystal arrays: (**d**) optical microscope images of TIPS-PEN crystals of different widths growing on a PUA/SiO_2_ substrate. (**e**) POM images. (**f**) AFM image and (**g**) its section height distribution [37]. Copyright© 2019 American Chemical Society.

**Figure 10 nanomaterials-13-02087-f010:**
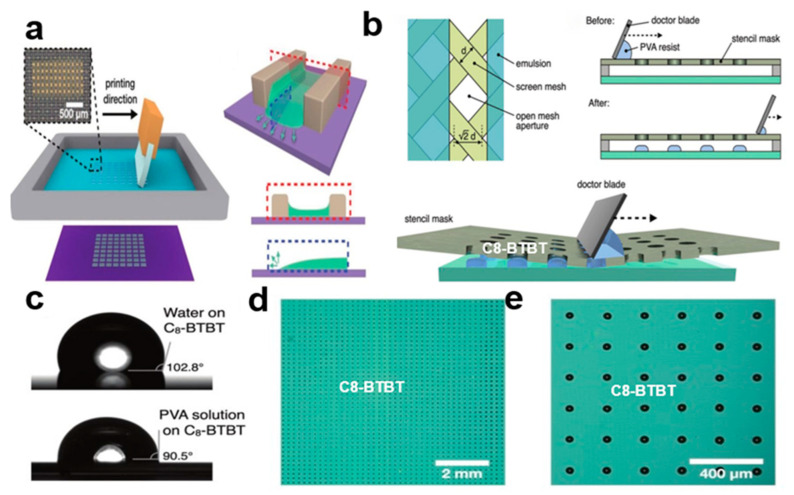
(**a**) Nanocrystal arrays created using screen printing [38]. Copyright© 2019 WILEY-VCH Verlag GmbH & Co. KGaA, Weinheim. (**b**) PVA resist stencil pattern printing [39]. Copyright© The Royal Society of Chemistry and the Chinese Chemical Society 2021. (**c**) Size of the contact angle of the C8-BTBT crystal array in water and PVA resist solution. (**d**) Optical microscope images of a C8-BTBT array with a 1 cm^2^ PVA template. (**e**) C8-BTBT array enlarged image.

**Figure 11 nanomaterials-13-02087-f011:**
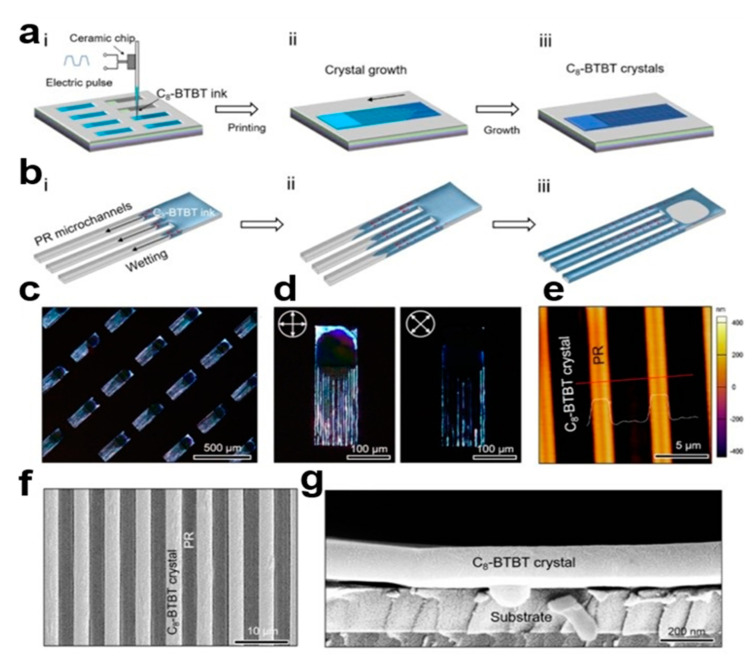
(**a**) Microchannel-assisted inkjet printing (MA-IJP) patterned C8-BTBT crystals. (**b**) COMSOL simulated solution wetting process. C8-BTBT crystal array: (**c**) Cross-polarization optical microscope (CPOM) image. (**d**) CPOM images of C8-BTBT crystals at 45° and 0° polarization angles. (**e**) AFM image. (**f**) SEM image [41]. Copyright© 2022 IOP Publishing Ltd. (**g**) SEM cross-section of C8-BTBT crystal.

**Figure 12 nanomaterials-13-02087-f012:**
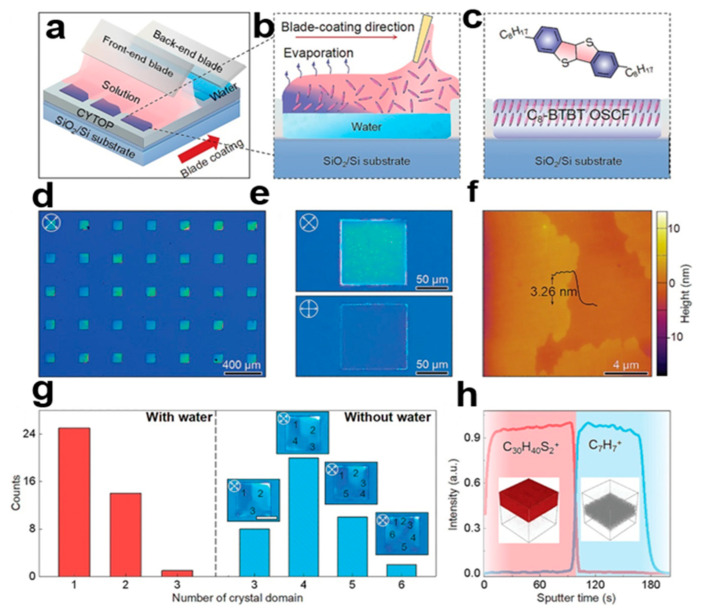
(**a**) C8-BTBT nanocrystals created using the double-blade coating technique [42]. Copyright© 2023 Wiley-VCH GmbH. (**b**) The back-end blue leaves the substrate wet with a small amount of water, and then the front-end pink blade coats the substrate with a mixture of organic small-molecule solution to grow the nanocrystal arrays. (**c**) Chemical structure of C8-BTBT. (**d**,**e**) C8-BTBT nanocrystal arrays with different polarization angles images. (**f**) Atomic force microscopy (AFM) image of C8-BTBT. (**g**) Statistical histogram showing the number of crystal domains. (**h**) Time-of-flight secondary ion mass spectrometry of mixed membrane C8-BTBT (TOF-SIMS).

**Figure 13 nanomaterials-13-02087-f013:**
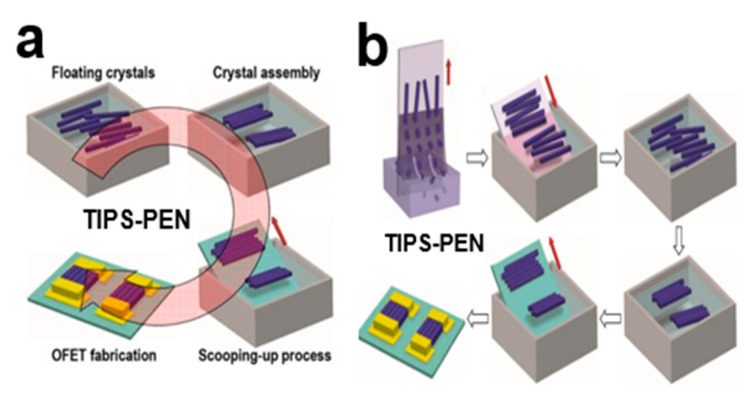
(**a**) Straightforward scooping-up of the TIPS-PEN nanocrystal arrays. (**b**) The SiO_2_ substrate was placed into the TIPS-PEN nanocrystal array and pulled upward by capillary action to obtain a self-stacking crystal band [44]. Copyright© 2016 American Chemical Society.

**Figure 14 nanomaterials-13-02087-f014:**
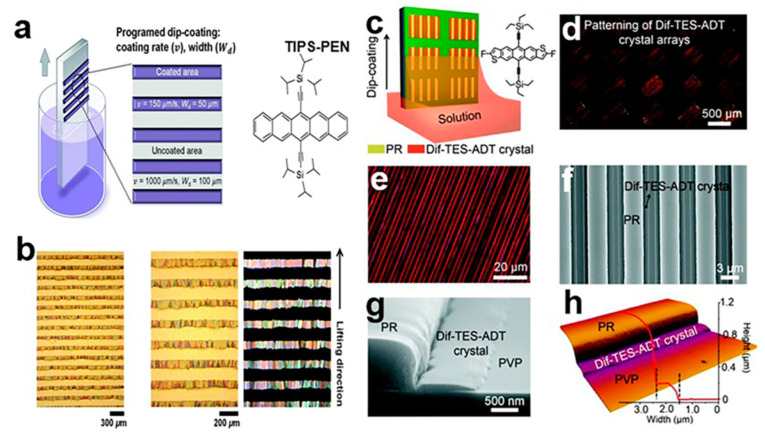
(**a**) Programmed dipping of TIPS-PEN nanocrystal arrays. (**b**) Optical microscope image of the TIPS-PEN fringe [45]. Copyright© Royal Society of Chemistry 2018. DIF-TES-ADT crystal array: (**c**) Patterned microchannel dip coating (PMDC) method. (**d**) Continuous CPOM image. (**e**) scaled CPOM image and (**f**) SEM image. (**g**) SEM cross-section image. (**h**) AFM image and height distribution [46]. Copyright© The Royal Society of Chemistry 2021.

**Figure 15 nanomaterials-13-02087-f015:**
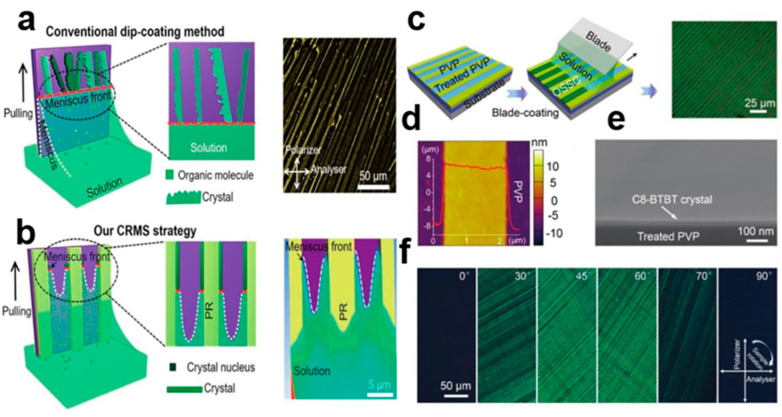
(**a**) Crystal growth by dipping on the meniscus [49]. Copyright© 2018 Elsevier Ltd. (**b**) C8-BTBT crystal arrays growth process in a channel-restricted meniscus. All rights reserved. C8-BTBT crystal array: (**c**) Blade coating on PVP surface [54]. Copyright© 2019 WILEY-VCH Verlag GmbH & Co. KGaA, Weinheim. (**d**) AFM image and height distribution. (**e**) SEM image. (**f**) CPOM images at different polarization angles: 0°, 30°, 45°, 60°, 70°, 90°.

**Figure 16 nanomaterials-13-02087-f016:**
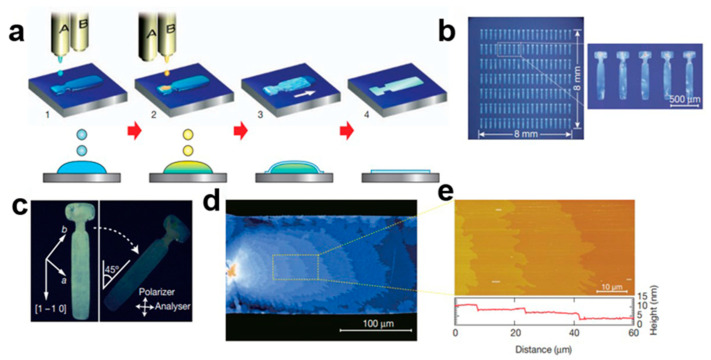
(**a**) Antisolvent ink (A) (step 1) is then sequentially overprinted with solution ink (B) to form a mixture of droplets confined to a predetermined area (step 2). Before the solvent evaporates completely (Step 4), the C8-BTBT crystal array grows at the interface (Step 3). (**b**) Image of C8-BTBT array for antisolvent crystallization. (**c**) Orthogonal Nichols polarizing optical micrograph of nanocrystal arrays. (**d**) Magnified micrograph of the nanocrystal arrays (**e**) AFM image of organic small-molecule nanocrystal arrays [55]. Copyright© 2011Macmillan Publishers Limited. All rights reserved.

**Figure 17 nanomaterials-13-02087-f017:**
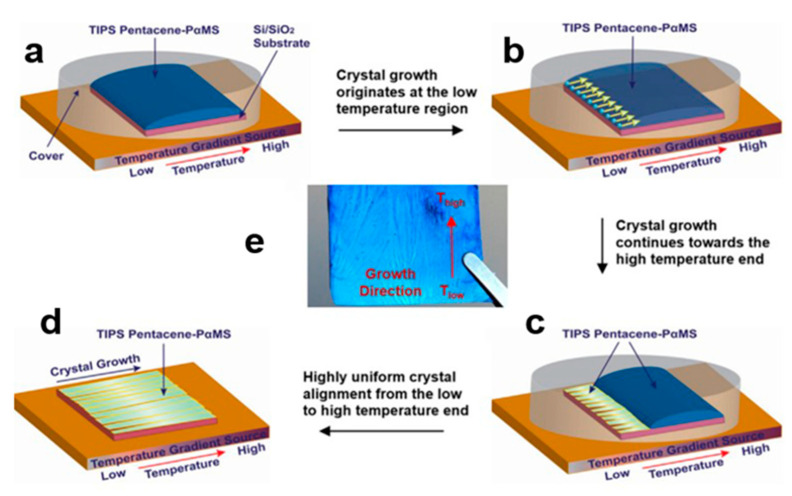
(**a**) Crystal growth device using the temperature gradient technique. (**b**) Crystal growth originates at the low temperature region. (**c**) Crystal growth continues towards the high temperature end. (**d**) Highly uniform crystal alignment from the low to high temperature end. (**e**) Images of TIPS-pentaphene crystals [61]. Copyright© 2016 Elsevier B.V. All rights reserved.

**Figure 18 nanomaterials-13-02087-f018:**
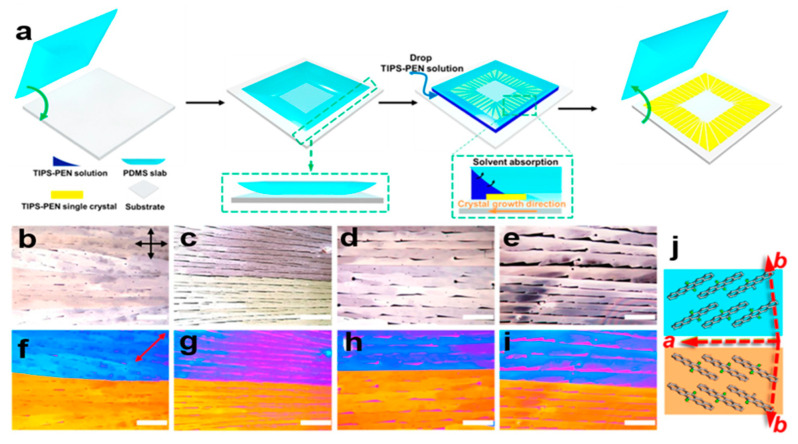
(**a**) Growth of TIPS-PEN nanocrystal array using the PDMS-assisted crystallization method. (**b**–**e**) Polarization optical microscope (POM) image of TIPS-PEN nanocrystal array with delay plate [64]. (**f**–**i**) POM image of a delay-free plate of a TIPS-PEN nanocrystal array. (**j**) Molecular orientation of TIPS-PEN nanocrystal arrays in blue and orange regions. Copyright© 2016 American Chemical Society.

**Figure 19 nanomaterials-13-02087-f019:**
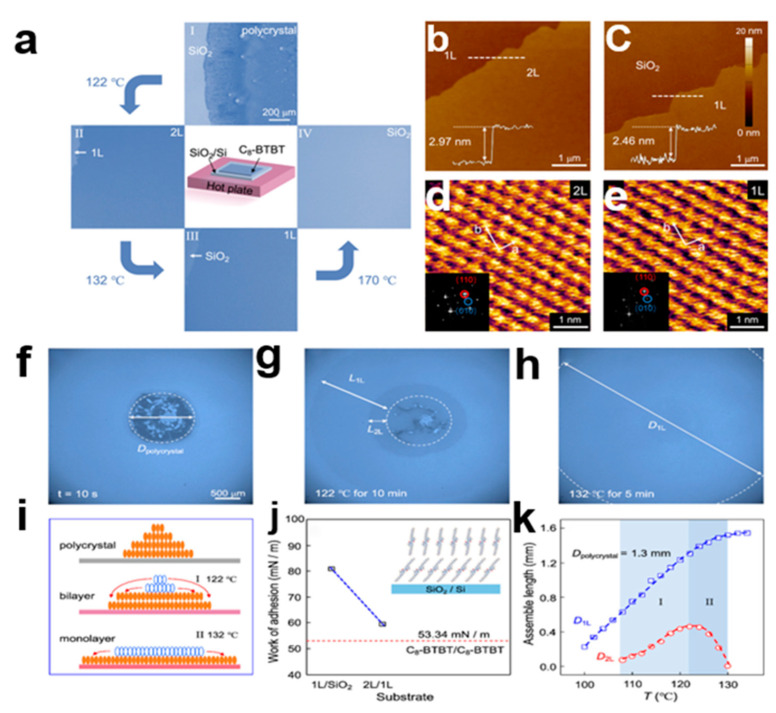
(**a**) Heat-induced self-assembly of C8-BTBT single crystal process [65]. Copyright© 2018 American Chemical Society. (**b**) AFM image of double-layer (2L) C8-BTBT crystal. (**c**) AFM images of single-layer (1L) C8-BTBT crystals. Scale: 1 μm. (**d**) HRAFM images of double-layer (2L) C8-BTBT crystal array on SiO_2_ substrate. (**e**) HRAFM images of single-layer (1L) C8-BTBT crystal array on SiO_2_ substrate. Scale: 1 nm. (**f**) Optical microscopy (OM) image of the initial crystal of C8-BTBT. (**g**) OM images of double-layer (2L) and single-layer (1L) C8-BTBT crystals annealed at 122 °C for 10 min. (**h**) OM image of C8-BTBT monolayer at 132 °C after annealing for 5 min. (**i**) The growth process from polycrystalline form to bilayer and monolayer C8-BTBT crystal arrays. (**j**) Molecular arrangement packing orientation between 1L C8-BTBT and SiO_2_, and between 2L C8-BTBT and 1L C8-BTBT. (**k**) The assembly length of 1L and 2L C8-BTBT crystals varies with substrate temperature (T).

**Figure 20 nanomaterials-13-02087-f020:**
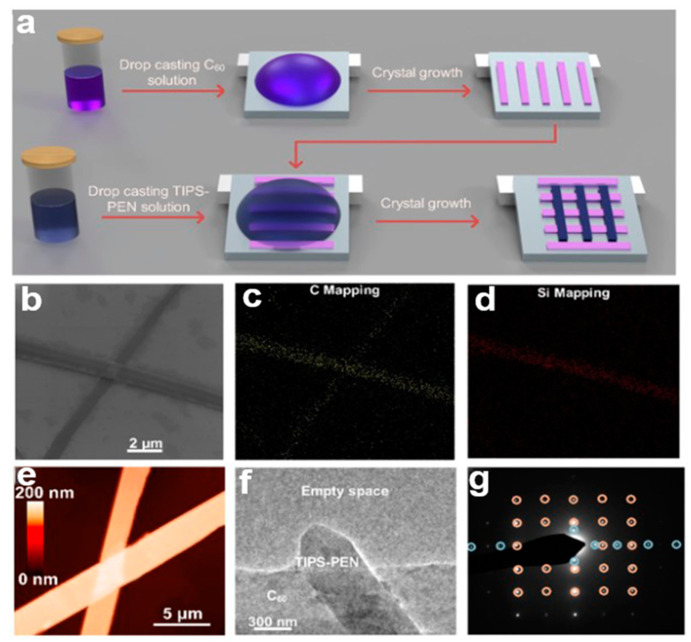
(**a**) Schematic diagram of a TIPS-PEN/C60 single-crystal heterojunction. (**b**) SEM image of an SCHJ. (**c**) Energy dispersive spectroscopy (EDS)-related element mapping. (**d**) EDS images. (**e**) AFM image. (**f**) TEM images of SCHJ. (**g**) Correlation electron diffraction pattern [68]. Copyright© 2018 American Chemical Society.

**Figure 21 nanomaterials-13-02087-f021:**
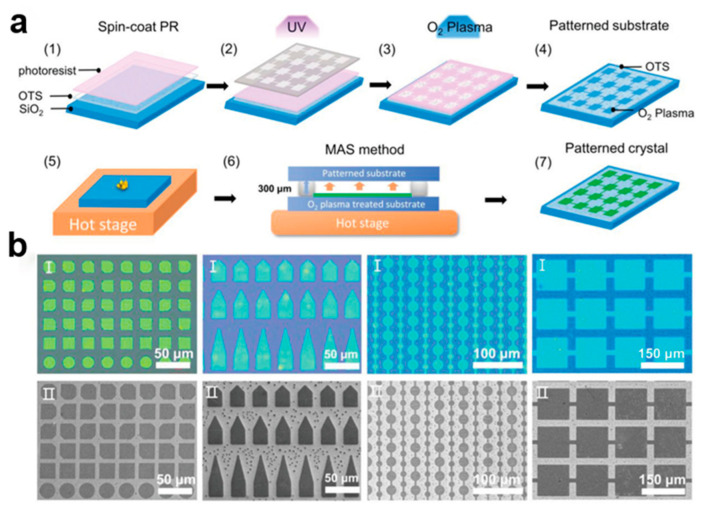
(**a**) Micro-spacing air sublimation grown C8-BTBT crystal arrays. (**b**) Optical microscope (OM) image (top) and scanning electron microscope (SEM) image (bottom) of a C8-BTBT single crystal with different sizes and shapes [72]. Copyright© 2023 Wiley-VCH GmbH.

**Figure 22 nanomaterials-13-02087-f022:**
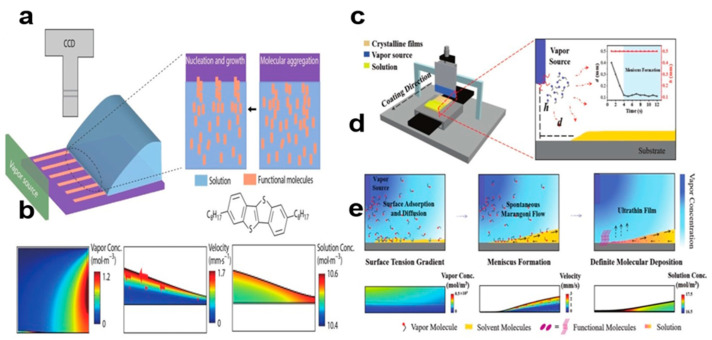
(**a**) Vapor-induced coating method of C8-BTBT crystals [74]. Copyright© 2023 Springer Nature Switzerland AG. Part of Springer Nature. (**b**) COMSOL simulates the process of vapor deposition of C8-BTBT crystals. (**c**) Single-crystal C8-BTBT created using a vapor coating device [75]. Copyright© 2023 Wiley-VCH GmbH. (**d**) Different stages of vapor-induced coating. (**e**) COMSOL simulated C8-BTBT vapor molecular distribution, velocity distribution, and solute concentration in the meniscus.

**Figure 23 nanomaterials-13-02087-f023:**
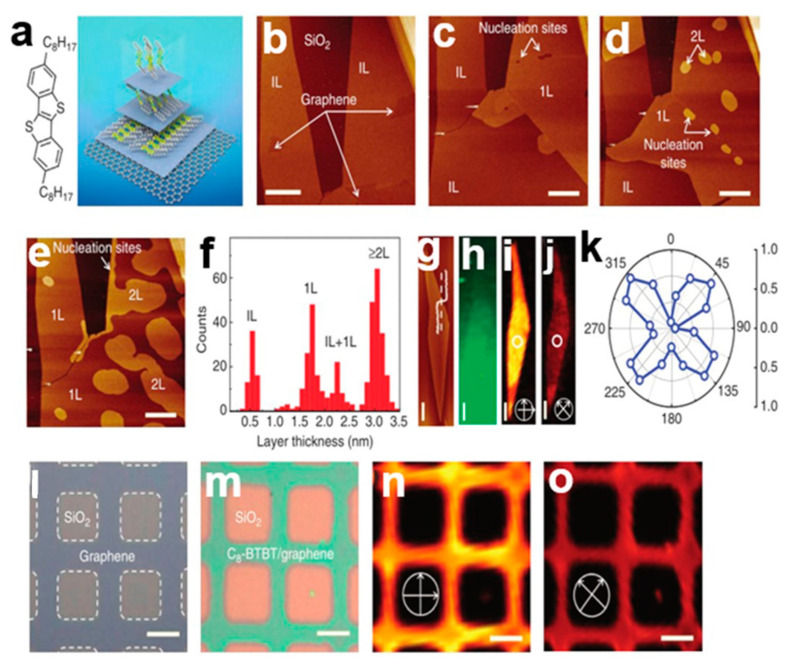
(**a**) Structural arrangement of C8-BTBT. (**b**–**e**) Atomic force microscope (AFM) images of C8-BTBT crystal at different growth stages. (**f**) Histogram of layer thickness of multiple samples of C8-BTBT crystal. (**g**) Atomic force microscope (AFM) image of C8-BTBT crystal array. (**h**) Raman mapping. (**i**,**j**) Cross-polarization optical micrograph of C8-BTBT crystal array. (**k**) Normalized strength of C8-BTBT crystal as a function of rotation angle. (**l**,**m**) Optical microscope image of C8-BTBT crystal growth pattern. (**n**,**o**) Cross-polarization optical micrograph of C8-BTBT growing at different polarization angles in the same region [77]. Copyright©2014 Macmillan Publishers Limited. All rights reserved.

**Figure 24 nanomaterials-13-02087-f024:**
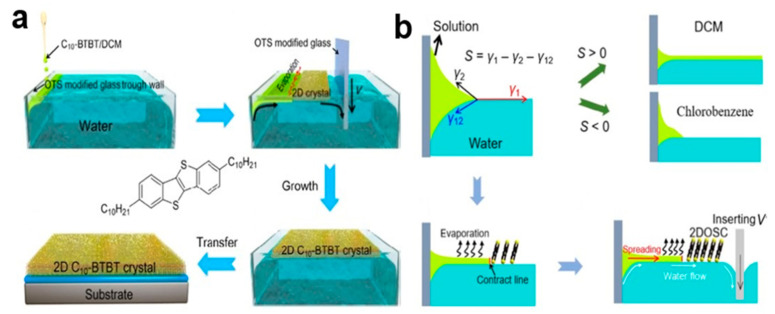
(**a**) EFDSE method for C10-BTBT 2D organic single crystals (2DOSCs) [81]. Copyright© Tsinghua University Press and Springer-Verlag GmbH Germany, part of Springer Nature 2019. (**b**) Growth mechanism of high—arranged large area 2DOSC. The 2DOSC C10-BTBT crystal array is subject to the combined action of three surface tensions.

**Figure 25 nanomaterials-13-02087-f025:**
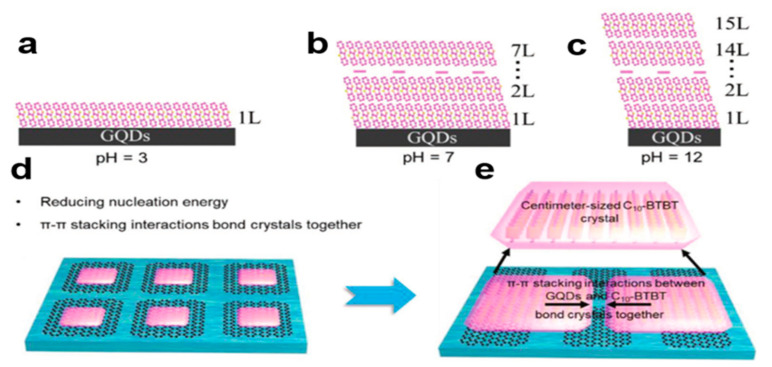
(**a**) pH = 3, C8-BTBT crystals can be stacked in a single layer in graphene quantum dot solution. (**b**) pH = 7, C8-BTBT crystals can be stacked in graphene quantum dot solution from 1 to 7 layers. (**c**) pH = 12, C8-BTBT crystals can be stacked in graphene quantum dot solution from 1 to 12 layers. (**d**,**e**) Quantum dots can reduce the intermolecular force of crystals and promote large area monolayer crystal aggregation [83]. Copyright© 2020 Wiley-VCH GmbH.

**Figure 26 nanomaterials-13-02087-f026:**
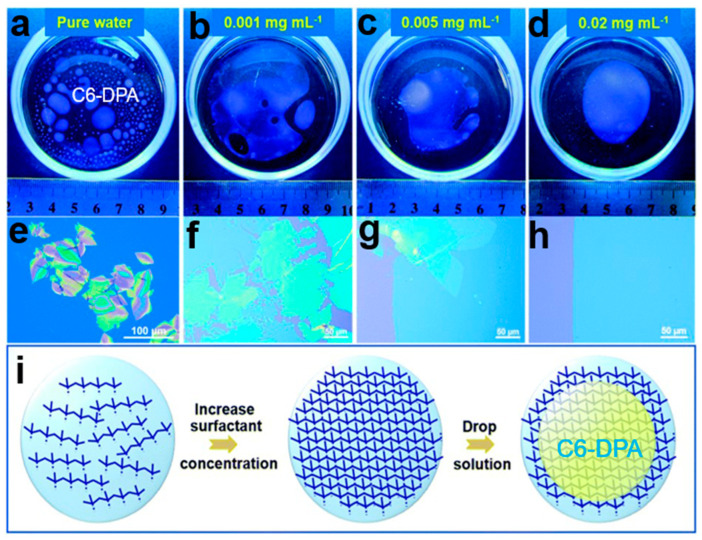
The growth of two-dimensional molecular crystals of C6-DPA at different concentrations of surfactant: (**a**) 0 mg mL^−1^, (**b**) 0.001 mg mL^−1^, (**c**) 0.005 mg mL^−1^, (**d**) 0.02 mg mL^−1^. (**e**–**h**) Optical microscope image of C10-BTBT crystal array grown with different concentrations of surfactants. (**i**) Soft template self-assembled C6-DPA grows two-dimensional organic single-crystal arrays [85]. Copyright© The Royal Society of Chemistry 2022.

**Figure 27 nanomaterials-13-02087-f027:**
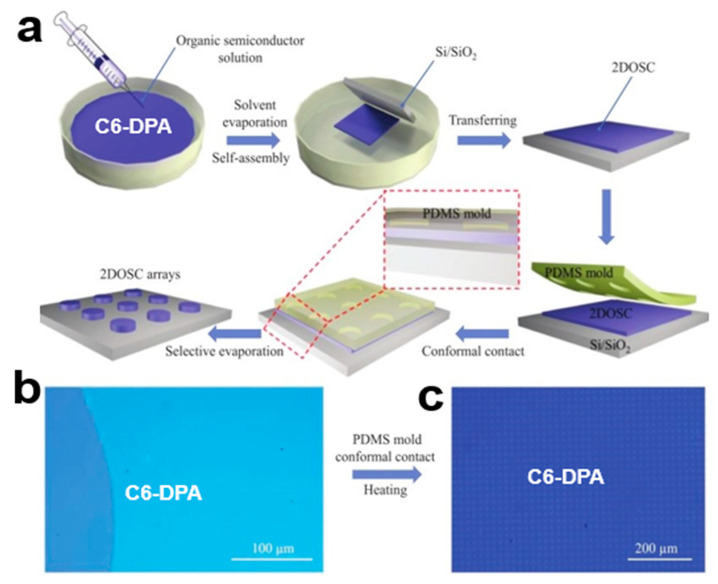
(**a**) Patterned layered 2DOSC array. (**b**) Large area C6-DPA 2DOSC and high-resolution optical microscope (OM) images; scale: 100 μm. (**c**) OM images of C6-DPA crystal arrays; scale: 200 μm [86]. Copyright© 2021 Wiley-VCH GmbH.

**Figure 28 nanomaterials-13-02087-f028:**
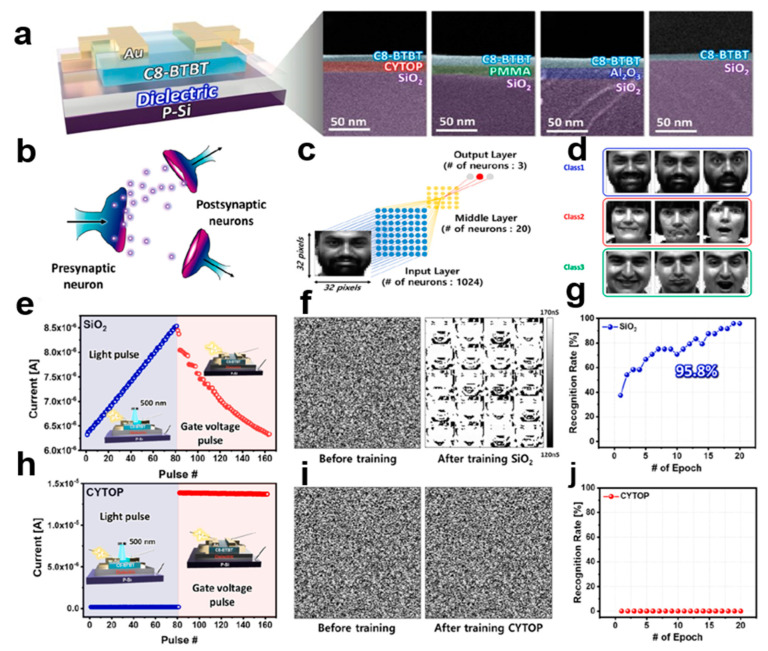
(**a**) A synaptic phototransistor composed of C8-BTBT nanocrystal array and dielectric layer and its cross-sectional SEM images. (**b**) A biological synapse composed of neurons, neurotransmitters, and postsynaptic neurons. (**c**) An artificial synapse composed of an organic small molecule C10-BTBT nanocrystal array. (**d**) Nine 32 × 32 pixel trained face image samples extracted from the face database. OPT based on C8-BTBT crystal array SiO_2_: (**e**) Current curve and recognition accuracy curve of artificial synaptic simulated face recognition. (**f**) Relationship between face recognition and training in artificial synaptic network. (**g**) Face recognition rate after training. OPT of crystal array CYTOP based on C8-BTBT: (**h**) Current curve and recognition accuracy curve of artificial synaptic simulated face recognition. (**i**) Relationship between face recognition and training in artificial synaptic network. (**j**) Face recognition rate after training [89]. Copyright© 2020 Elsevier Ltd. All rights reserved.

**Figure 29 nanomaterials-13-02087-f029:**
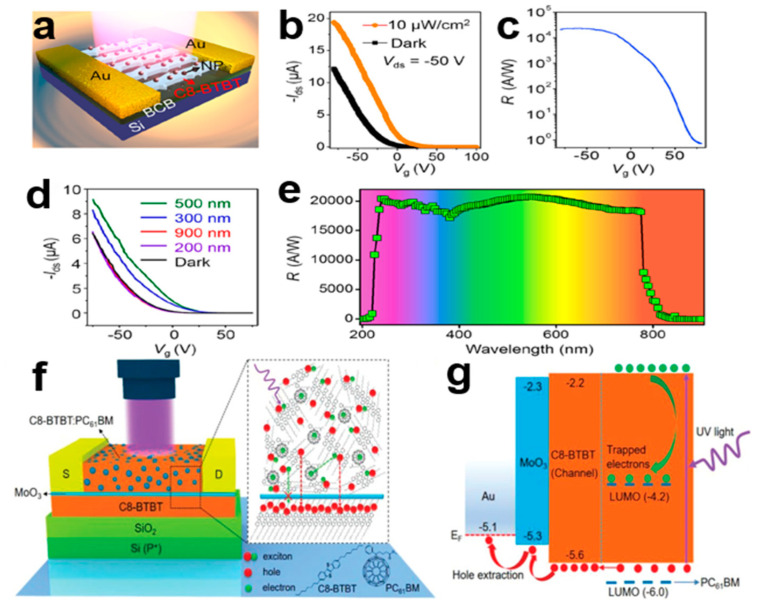
(**a**) Phototransistor composed of C8-BTBT single-crystal array. (**b**) Transmission characteristics of C8-BTBT phototransistors at dark and 550 nm illumination (10 μW/cm^2^). (**c**) Variation of photoresponse characteristics of perovskite NPs/C8-BTBT phototransistor with $V_{g}$. (**d**) Transmission characteristics of NPs/C8-BTBT hybrid transistors at dark and different wavelengths [95]. (**e**) Optical responsiveness as a function of optical wavelength [96]. Copyright©2019 WILEY-VCH Verlag GmbH & Co. KGaA, Weinheim. (**f**) Structure diagram of UV-triggered HL-OPT and photoelectric detection diagram of C8-BTBT crystal array. (**g**) Energy level diagram of HL-OPT device under ultraviolet irradiation.

**Figure 30 nanomaterials-13-02087-f030:**
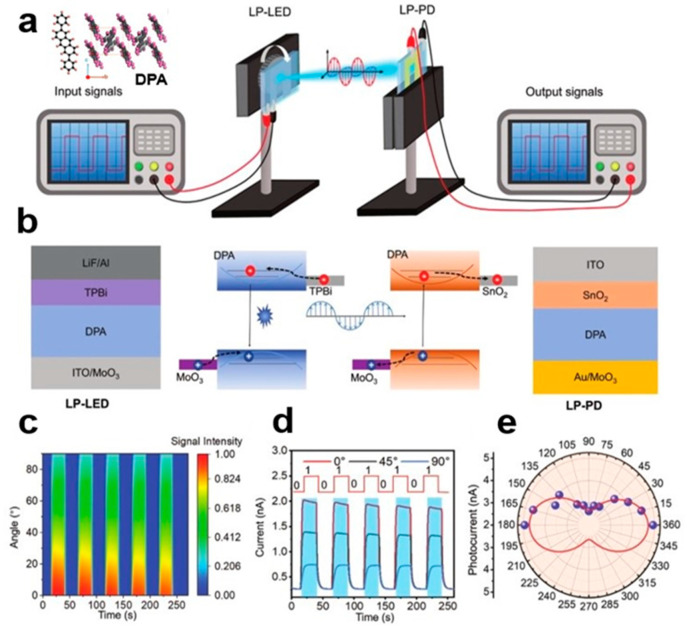
(**a**) DPA SC inter-chip polarized optical communication system. (**b**) Operating mechanism of LP-LED and LP-PD devices and integrated systems. (**c**) Normalized polarization depends on the output signal strength and the input signal period of the isogram. (**d**) Repeatable optical response signals of LP-PD at 0°, 45°, and 90° polarization angles. The blue shade indicates the time of the polarized light. (**e**) Polaroid diagram of photocurrent and polarization angle and fitting curve [97]. Copyright© 2023 Wiley-VCH GmbH.

**Figure 31 nanomaterials-13-02087-f031:**
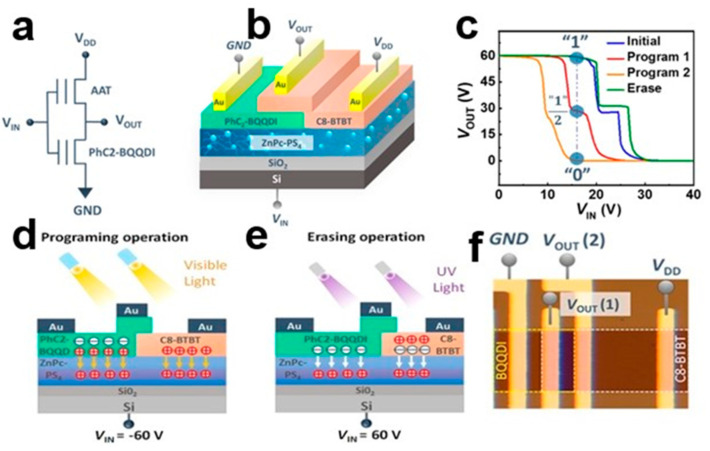
(**a**) Ternary inverter circuit diagram. (**b**) Schematic diagram of three-way inverter composed of anti-bipolar transistor (AAT) and PhC2-BQQDI transistor. (**c**) Optical programming and erasure operation of ternary storage inverters. (**d**) Inverter visible-light-assisted programming operation schematic and energy level diagram. (**e**) Inverter UV-assisted erase operation schematic and energy level diagram. (**f**) Inverter optical microscope image [98]. Copyright© 2022 American Chemical Society.

## Data Availability

Not applicable.

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
