# Peer review of "Nanocrystal Array Engineering and Optoelectronic Applications of Organic Small-Molecule Semiconductors"

_nanomaterials, 2023, doi:10.3390/nano13142087_

Round 1
Reviewer 1 Report
The review is focused on small-molecule organic semiconductors, particularly preparation techniques.
Next, the authors highlight the “organic small-molecules semiconductors”, but their relation to other types semiconductors is not clear. Are there organic semiconductors that are not based on small molecules? Please consider adding 1-2 sentences to the introduction that define and narrow the subject.
Unfortunately, the main part is not very useful. Sections are too short and do not provide general information about methods. Generally, each section consists mainly of one particular figure, either combined or simple repeated from previously published papers, with small text description paraphrasing main results of the paper(s). Sometimes not all panels of figures are described (for instance, Figure 23 with 15 different images is described as follows: ”(a-o) Growth and characterization of C8-BTBT single crystals on graphene by van der Waals force epitaxy[77]”). So, the manuscript is just a collection of sources, and does not analyze, synthesize, and critically evaluate them to give the whole picture.
Recommended change in the title: the phrase “organic small molecular semiconductors” is quite unusual, consider using “organic small-molecule semiconductors” or “small-molecule organic semiconductors”.
The same in abstract (“organic small molecular nanocrystal array film preparation methods”). This sentence is also quite long, can it be simplified?
Different terms are used in the paper: sometimes the authors use “nanocrystal array”, sometimes “nanocrystal array film” and sometimes “nanocrystal films”. If these terms have the same meaning, it is better to stick to one of them. If not, please explain the difference.
Please define abbreviations: OLED, TIPS-PEN, OM, TIPS-TAP and many others.
Author Response
Comments to the Author
The review is focused on small-molecule organic semiconductors, particularly preparation techniques.
Next, the authors highlight the “organic small-molecules semiconductors”, but their relation to other types semiconductors is not clear. Are there organic semiconductors that are not based on small molecules? Please consider adding 1-2 sentences to the introduction that define and narrow the subject.
Unfortunately, the main part is not very useful. Sections are too short and do not provide general information about methods. Generally, each section consists mainly of one particular figure. Either
Combined or simple repeated form previously published papers, with small text description paraphrasing main results of the paper(s).Sometimes not all panels of figures are described (for instance, Figure 23 with 15 different images is described as follows:“(a-o) Growth and characterization of C8-BTBT single crystals on graphene by van der Waals force epitaxy[77])”).So, the manuscript is just a collection of sources, and does not analyze, synthesize, and critically evaluate them to give the whole picture.
1. Next, the authors highlight the “organic small-molecules semiconductors”, but their relation to other types semiconductors is not clear. Are there organic semiconductors that are not based on small molecules? Please consider adding 1-2 sentences to the introduction that define and narrow the subject.
Response: We appreciate the referee’s suggestion. In revised manuscript we have narrowed down the field by explaining the organic small molecule semiconductors. Organic small molecule semiconductors are very small molecular weight organic molecules, their molecular weight is only a few hundred to a few thousand, can be used as a hole transport layer in optoelectronic devices, with high electrical properties. In this paper, the new progress in the engineering of nanocrystalline arrays based on organic small molecule semiconductors is reviewed, and the advantages and disadvantages of preparation methods of organic small molecule semiconductors are summarized. Several common organic small molecule semiconductors, such as pentaphenes and their derivatives, benzothiophene groups and 2, 6-bis (4-hexylphenyl) anthracene (C6-DPA) etc. It is illustrated. They are composed by adding different alkyl chains to the side chain on the basis of small organic molecule groups. The methods in each chapter are applicable to the preparation of all organic small molecule crystals, not only to a particular kind of organic small molecule, but also to this class of organic small molecule. This paper reviews the organic small molecule semiconductors. All the ones listed in this paper are organic small molecule semiconductors, from its picture chemical structure and chemical name are easy to know are organic small molecular semiconductors. there are no other kinds of semiconductors. Most of the literatures used in this paper are from the last five years, and the methods summarized are all new. We have revised and supplemented each chapter. We highlight the revised chapters in yellow. Thank you again for your positive and constructive comments and suggestions on our manuscript.
2. the main part is not very useful. Sections are too short and do not provide general information about methods. Generally, each section consists mainly of one particular figure. Either Combined or simple repeated form previously published papers, with small text description paraphrasing main results of the paper(s).
Response: We appreciate the referee’s suggestion. In revised manuscript we have made changes to each section. We have also added to the content of each section. The chapters of the main section have been supplemented with a lot of content, and the method is summarized in detail in small text. We highlight the revised chapters in yellow. Thank you again for your positive and constructive comments and suggestions on our manuscript.
3. Sometimes not all panels of figures are described (for instance, Figure 23 with 15 different images is described as follows:“(a-o) Growth and characterization of C8-BTBT single crystals on graphene by van der Waals force epitaxy[77])”).So, the manuscript is just a collection of sources, and does not analyze, synthesize, and critically evaluate them to give the whole picture.
Response: We appreciate the referee’s suggestion. In revised manuscript We have modified the picture, given a specific description, and conducted an analysis to comprehensively evaluate the overall picture of the method. (For example, there are 15 different images in Figure 23, and Figure [77]"). We highlight the revised chapters in yellow. Thank you again for your positive and constructive comments and suggestions on our manuscript.
4. Recommended change in the title: the phrase“organic small molecular semiconductors” is quite unusual consider using“organic small-molecular semiconductors” or “small-molecular organic semiconductors” .
The same in abstract(“organic small molecular nanocrystal array film preparation methods”).This sentence is also quite long, can it be simplified?
Different terms are used in the paper: sometimes the authors use “nanocrystal array”,sometimes “nanocrystal array film”and sometimes“nanocrystal film”,If these terms have the same meaning, it is better to stick to one of them. If not, please explain the difference.
Please define abbreviations: OLED,TIPS-PEN,OM,TIPS-TAP and many others.
Response: We appreciate the referee’s suggestion. In the revised manuscript ,the title we have been changed to "organic small-molecule semiconductor", abstract that sentence can be simplified to “nanocrystal array engineering”, different terms are used in the paper: sometimes "nanocrystal array" is used, sometimes "nanocrystal array thin film" is used, sometimes "nanocrystal thin film" is used, these terms have the same meaning, we have decided to use “nanocrystal arrays”. We highlight the revised chapters in yellow. Thank you again for your positive and constructive comments and suggestions on our manuscript.
Response: We appreciate the referee’s suggestion. In the revised manuscript we have added the following description of the definition of acronyms; OLED, OM image, TIPS-PEN, Dif-TES-ADT,C8-BTBT,PDMS,PMMA,PUA mode, etc.
An acronym for instrumental tests and indicators :OLED (Organic Light-Emitting Diode), light-emitting diodes (LP-LEDs). liquid-insulator-bridging (LIB) .OM image(Optical microscope image). Atomic force microscopy (AFM) , Scanning Electron Microscope(SEM), Cross polarization optical microscope (CPOM), Atomic Force Microscope(AFM), energy dispersive spectroscopy (EDS), Transmission electron microscopy(TEM), select area electron diffraction (SAED)
(2D) organic single crystals (2DOSCs)
External force driven solution epitaxy (EFDSE)
These abbreviations are organic small molecule semiconductors:
TIPS-PEN[6,13-bis(triisopropylsilylethynyl)pentacene],
2,6-bis(4-hexylphenyl)anthracene (C6-DPA)
Dif-TES-ADT[2,8-difluoro-5,11-bis(triethylsilylethynyl)anthradithiophene],
C10-BTBT[2,7-didecylbenzothienobenzothiophene]
OTS[octadecyltrichlorosilane]
C8-BTBT(2,7-dioctyl[1]benzothieno[3,2-b] benzothiophene). etc
Material used for substrate:
Polyvinyl alcohol (PVA)
poly(4-vinylphenol) (PVP)
PDMS(Polydimethylsiloxane)
PMMA(Polymethyl methacrylate)
PUA (polyurethane acrylate)
These abbreviations refer to organic small molecule semiconductor substances and the templates used to grow them in nanocrystals. These methods in the original literature are applicable to many small organic molecules. Some common small organic molecules are introduced as examples.
Abbreviations for these methods can be seen in the title of each chapter, or their chemical structure can be seen in the picture. The full names of these abbreviations have been supplemented in the text. We highlight the revised chapters in yellow. Thank you again for your positive and constructive comments and suggestions on our manuscript.
Reviewer 2 Report
Report on the paper by Gong et al.
Although it is an important review article on various methods of microstructural template patterning of nanocrystals of organic semiconductors, there are some unclear points.
1. Typical size of nanocrystals should be discussed for each method.
2. Possibility of mesocrystal formation in the nanocrystal array engineering should be discussed in the manuscript.
3. Definitions of the following abbreviations should be explained in the manuscript; OLED, OM images, TIPS-PEN, Dif-TES-ADT, C8-BTBT, PDMS, PMMA, PUA pattern, etc.
4. The following terms should be explained in more details in the manuscript; scraper method, grove coting method, etc.
Some revision is required.
Author Response
Comments to the Author
Although it is an important review article on various methods of microstructural template patterning of nanocrystals of organic semiconductors, there are some unclear points.
- Typical size of nanocrystals should be discussed for each method.
Response: We appreciate the referee’s suggestion. In revised manuscript we have added a typical size description of the nanocrystals for each method. The size of the organic small molecule nanocrystal array is between 1 and 10, and its size is related to the size of the specific photoelectric device. We highlight the revised chapters in yellow. Thank you again for your positive and constructive comments and suggestions on our manuscript.
- Possibility of mesocrystal formation in the nanocrystal array engineering should be discussed in the manuscript.
Response: We appreciate the referee’s suggestion. In the revised manuscript we have added the possibility of the formation of engineered mesocrystal in nanocrystalline arrays. We highlight the revised chapters in yellow. Thank you again for your positive and constructive comments and suggestions on our manuscript.
- Definitions of the following abbreviations should be explained in the manuscript; OLED, OM images, TIPS-PEN, Dif-TES -ADT,C8-BTBT,PDMS,PMMA,PUA pattern, etc.
Response: We appreciate the referee’s suggestion. In the revised manuscript we have added the following description of the definition of acronyms; OLED, OM image, TIPS-PEN, Dif-TES-ADT,C8-BTBT,PDMS,PMMA,PUA mode, etc.
An acronym for instrumental tests and indicators :OLED (Organic Light-Emitting Diode), OM image(Optical microscope image). Atomic force microscopy (AFM) , Scanning Electron Microscope(SEM), Cross polarization optical microscope (CPOM), Atomic Force Microscope(AFM), energy dispersive spectroscopy (EDS), Transmission electron microscopy(TEM), select area electron diffraction (SAED)
(2D) organic single crystals (2DOSCs), Two-dimensional molecular crystal (2DMC), organic phototransistors (OPTs) ,octadecyltrichlorosilane (OTS)
External force driven solution epitaxy (EFDSE)
These abbreviations are organic small molecule semiconductors:
TIPS-PEN[6,13-bis(triisopropylsilylethynyl)pentacene],
2,6-bis(4-hexylphenyl)anthracene (C6-DPA)
Dif-TES-ADT[2,8-difluoro-5,11-bis(triethylsilylethynyl)anthradithiophene],
C10-BTBT[2,7-didecylbenzothienobenzothiophene]
OTS[octadecyltrichlorosilane]
C8-BTBT(2,7-dioctyl[1]benzothieno[3,2-b] benzothiophene).
PhC2H4-benzo[de]isoquinolino[1,8-gh]quinolone diimide (PhC2-BQQDI) etc
Material used for substrate:
Polyvinyl alcohol (PVA)
poly(4-vinylphenol) (PVP)
PDMS(Polydimethylsiloxane)
PMMA(Polymethyl methacrylate)
PUA (polyurethane acrylate)
These abbreviations refer to organic small molecule semiconductor substances and the templates used to grow them in nanocrystals. These methods in the original literature are applicable to many small organic molecules. Some common small organic molecules are introduced as examples.
Abbreviations for these methods can be seen in the title of each chapter, or their chemical structure can be seen in the picture. The full names of these abbreviations have been supplemented in the text. We highlight the revised chapters in yellow. Thank you again for your positive and constructive comments and suggestions on our manuscript.
- The following terms should be explained in more details in the manuscript; scraper method, grove coating method, etc
Response: We appreciate the referee’s suggestion. In revised manuscript we have added the detailed elaboration of scraper method, grating coating method, etc. We highlight the revised chapters in yellow. Thank you again for your positive and constructive comments and suggestions on our manuscript.
Round 2
Reviewer 1 Report
The manuscript was improved after revision, but still there are some questions.
1. The authors claim that the title we have been changed to "organic small-molecule semiconductor". But in the manuscript it is "organic small-molecular semiconductors ", which is less common. I recommend to use "organic small-molecule semiconductor" in the title and throughout the paper.
2. The new sentence that has been added to abstract is gramatically incorrect: "Organic small-molecular semiconductors are very small molecular-weight, its molecular-weight is only a few hundred to a few thousand, can be used as a hole transport layer in optoelectronic devices, with high electrical properties.". Also, the meaning of the phrase "high electrical properties" is not clear.
3. The relation of small-molecule organic semiconductors to other types of semiconductors is still not clear. Please add the motivation to the introduction: (a) why to study small-molecule organic semiconductors, in what terms they are better than other types? and (b) for what purpose one need to prepare films/array of them?
4. The authors responded that the sentense in the abstract "organic small-molecular nanocrystal array film preparation methods" can be simplified to "nanocrystal array engineering", but it is still the same in the manuscript.
-
Author Response
Comments and Suggestions for Authors
The manuscript was improved after revision, but still there are some questions.
1. The authors claim that the title we have been changed to "organic small-molecule semiconductor". But in the manuscript it is "organic small-molecular semiconductors ", which is less common. I recommend to use "organic small-molecule semiconductor" in the title and throughout the paper.
Answer: We appreciate the referee’s suggestion. In revised manuscript we have used "organic small-molecule semiconductor" in the title and throughout the paper. We highlight the revised chapters in red. Thank you again for your positive and constructive comments and suggestions on our manuscript.
2. The new sentence that has been added to abstract is gramatically incorrect: "Organic small-molecular semiconductors are very small molecular-weight, its molecular-weight is only a few hundred to a few thousand, can be used as a hole transport layer in optoelectronic devices, with high electrical properties.". Also, the meaning of the phrase "high electrical properties" is not clear.
Answer: We appreciate the referee’s suggestion. In revised manuscript we have modified that “Compared to organic macromolecule, organic small molecule exhibit better crystallinity and therefore show better semiconductor performance.”. We highlight the revised chapters in red. Thank you again for your positive and constructive comments and suggestions on our manuscript.
3. The relation of small-molecule organic semiconductors to other types of semiconductors is still not clear. Please add the motivation to the introduction: (a) why to study small-molecule organic semiconductors, in what terms they are better than other types? and (b) for what purpose one need to prepare films/array of them?
Answer: We appreciate the referee’s suggestion. In the revised introduction part, we have addressed above issues by adding new discription “On the one hand, organic small-molecule semiconductor materials typically exhibit better semiconductor performance than polymer semiconductor materials due to their better crystallization properties. On the other hand, compared to traditional silicon-based semiconductor materials, organic small-molecule semiconductor materials also have the characteristic of being solution processable. Therefore, it is more suitable for the low energy consumption needs of future flexible electronic devices. In addition, the preparation of high-quality and high-density crystal arrays is crucial for the practical application of organic small molecule semiconductor materials. Taking information display as an example, the density of organic small-molecule crystal array determines the pixel density. Therefore, research on the processing technology of organic small molecule semiconductor has always been a hotspot of organic electronics”. Thank you again for your positive and constructive comments and suggestions on our manuscript.
4. The authors responded that the sentence in the abstract "organic small-molecular nanocrystal array film preparation methods" can be simplified to "nanocrystal array engineering", but it is still the same in the manuscript.
Answer: We appreciate the referee’s suggestion. In revised manuscript we have simplified the abstract "preparation method of organic small molecule nanocrystalline array thin film" to "nanocrystalline array engineering". We highlight the revised chapters in red. Thank you again for your positive and constructive comments and suggestions on our manuscript.